



# A reactive nitrogen budget of the Bohai Sea based on an isotope mass balance model

Shichao Tian[1], Birgit Gaye[1], Jianhui Tang[2], Yongming Luo[2, 3,] Wenguo Li[4], Niko Lahajnar[1], Kirstin Dähnke[5], Tina Sanders[5], Tianqi Xiong[6], Weidong Zhai[6], Kay-Christian Emeis[1, 5]

5   [1]Institute for Geology, Universität Hamburg, Hamburg, 20146, Germany

[2]Yantai Institute of Coastal Zone Research, Chinese Academy of Sciences, Yantai, China

[3]Institute of Soil Science, Chinese Academy of Siences, Nanjing, China

[4]Institute of Oceanography, Universität Hamburg, Hamburg, 20146, Germany

[5]Helmholtz-Zentrum Geesthacht (HZG), Institute for Coastal Research, Geesthacht, 21502, Germany

10   [6]Institute of Marine Science and Technology, Shandong University, Qingdao, China

*Correspondence to*: Shichao Tian (shichao.tian@uni-hamburg.de)



**Abstract.** The Bohai Sea is a semi-closed marginal sea impacted by one of the most populated areas of China. The supply of nutrients, markedly that of reactive nitrogen, via fluvial and atmospheric transport has strongly increased in parallel with the growing population. It is therefore crucial to quantify the reactive nitrogen input to the BHS and to understand the processes and determine the quantities of nitrogen eliminated in and exported from the BHS. The nitrogen budget and in particular the internal sources and sinks of nitrate were constrained by using a mass-based and dual stable-isotope approach based on $\delta^{15}N$ and $\delta^{18}O$ of nitrate. Samples of water, suspended matter and sediments were taken in the BHS in spring (March and April) and summer (July and August) 2018. The Yellow River was sampled in May, July to November and Daliao River, Hai River, Luan River and Xiaoqing River were sampled in November of 2018. In addition to nutrient, particulate organic carbon and nitrogen concentrations, the dual isotopes of nitrate ($\delta^{15}N$ and $\delta^{18}O$), $\delta^{15}N$ of suspended matters and sediments were determined. Based on the available mass fluxes and isotope data an updated nitrogen budget is proposed. Compared to previous estimates, it is more complete and includes the impact of interior cycling (nitrification) on the nitrate pool. The main nitrogen sources are rivers contributing 17.5%-20.6% and the combined terrestrial runoff (including submarine discharge of nitrate with fresh ground water) accounting for 22.6%-26.5% of the nitrate input to the BHS while atmospheric input contributes only 6.3%-7.4% to total nitrate. An unusually active interior nitrogen cycling contributes 59.1%-71.2% to total nitrate via nitrification. Nitrogen is mainly trapped in the BHS and mainly removed by sedimentation (96.4%-96.9%) and only very little is exported to the YS (only 1.7%-2.0%). At present denitrification is only active in the sediments and removes 1.4%-1.7% of nitrate from the pool. A further eutrophication of the BHS could, however, induce water column hypoxia and denitrification as already observed – often seasonally off river mouths - in other marginal seas.



## 1. Introduction

Reactive or fixed nitrogen ($N_r$) is an essential nutrient of life on earth, but only few organisms can fix dinitrogen directly from the large atmospheric dinitrogen pool. Since the invention of the Haber-Bosch process the amount of fixed nitrogen (120

Tg N yr$^{-1}$) has constantly grown and since 2010 exceeds the natural terrestrial sources of reactive N of 63Tg N yr$^{-1}$ (Fowler et al., 2013). Hotspots of agricultural N fertilizer application shifted from the US and western Europe in the 1960s to eastern Asia in the early 21$^{st}$ century (Lu and Tian, 2017). In China, the Haber-Bosch process produces 37.1 Tg N yr$^{-1}$, which is almost 3 times of the biological N fixation of 12.0 Tg N yr$^{-1}$. An estimated 32.0 Tg N yr$^{-1}$ are produced for fertilizer (Gu et al., 2015) and China accounted for 29% of the global ammonium production in 2018 (IFA, 2019). The leakage and volatilization of this

man-made reactive nitrogen has strongly impacted limnic and marine ecosystems in China. Riverine reactive nitrogen discharged to the ocean from China was estimated at 5.4 Tg N yr$^{-1}$ (Gu et al., 2015). The total load of Chinese major estuaries to coastal seas was about 9% of the global river load for DIN and 1.5% of the global phosphate load, respectively (Smith et al., 2003;Liu et al., 2009).

The Bohai Sea (BHS) is a semi-enclosed basin with a surface area of $77\times10^3$km$^2$ and an average depth of 18 m (Chen,

2009;Su, 2001) that is heavily impacted by human activities in one of the most densely populated terrestrial catchments of the world. It exchanges salt water with the Yellow Sea (YS) through Bohai Strait and the Yellow River is a major source of freshwater to BHS (Chen, 2009). During the last fifty years, rising anthropogenic activity in the catchment induced severe environmental changes in the BHS, including increasing salinity, temperature, concentrations of dissolved inorganic nitrogen (DIN) and changes in stoichiometric nutrient ratios (Zhao et al., 2002;Zhang et al., 2004;Wang et al., 2019;Ning et al., 2010).

DIN concentrations increased from 0.30µmol L$^{-1}$ to 3.55µmol L$^{-1}$ in the time from 1982-2009, while phosphate (from 0.76µmol L$^{-1}$ to 0.31µmol L$^{-1}$) and silicate (26.6µmol L$^{-1}$ to 6.60µmol L$^{-1}$) concentrations significantly decreased, so that N/P increased dramatically (Zhang et al., 2004;Liu et al., 2011). Phytoplankton nutrient limitation in the BHS switched from nitrogen to phosphorus in the period of the 1980s to the 1990s and this limitation pattern persists until the present day (Xu et al., 2010;Liu et al., 2009;Wang et al., 2019).

The total annual water discharge of rivers into BHS is about $68.5\times10^9$m$^3$ yr$^{-1}$, of which the YR accounts for more than 75% (Liu et al., 2011). Water exchange time of the YR estuary is only 0.1-0.2 days (Liu et al., 2009), which implies a fast transfer of nutrients into the open BHS and much of these are trapped in Laizhou Bay (Fig. 1) (Zhang et al., 2004). The atmospheric deposition of nitrate ($3.42\times10^9$mol yr$^{-1}$) in BHS was modelled to be less than riverine nitrate ($7.25\times10^9$mol yr$^{-1}$), while more ammonium was supplied from atmospheric deposition ($6.15\times10^9$mol yr$^{-1}$) than from riverine input ($0.93\times10^9$mol

yr$^{-1}$) in the 1990s (Zhang et al., 2004). BHS nitrate budgets reported during the last two decades were not completely constrained, because crucial data, such as groundwater discharge or nitrification, were not available (Zhang et al., 2004;Liu et





al., 2003;Liu et al., 2009;Liu et al., 2011). There are few published nutrient data from the BHS over the last decade, and the terms in the $N_r$ budget of BHS concerning the quantities of $N_r$ generated or eliminated by biogeochemical cycling within the basin have not been addressed.

The nitrogen budget and in particular the internal sources and sinks of nitrate can be constrained with a mass-based and dual stable-isotope approach based on $\delta^{15}N$ and $\delta^{18}O$ of nitrate. The combination of data permits tracking of nitrate and quantification of internal cycling of inorganic nitrogen (Sigman et al., 2005;Wankel et al., 2006;Sugimoto et al., 2009;DiFiore et al., 2006;Montoya et al., 2002;Emeis et al., 2010). Stable isotopes of reactive nitrogen have been used to explore nitrogen sources in the eastern Chinese seas (Umezawa et al., 2013;Wang et al., 2016;Liu et al., 2017;Wu et al., 2019;Liu et al., 2020)
and the Changjiang Estuary (Yu et al., 2015;Wang et al., 2017;Yang et al., 2018;Chen et al., 2013).

For this study we analyzed water, suspended matter and sediments in the Bohai Sea sampled during the spring and summer seasons for nutrient concentrations, carbon and nitrogen contents, dual isotopes ($\delta^{15}N$ and $\delta^{18}O$) of nitrate and $\delta^{15}N$ of particulate nitrogen. Aim of the study is to characterize and quantify $N_r$ sources and sinks, in particular those from internal cycling processes that have not been included in previous budgets (Zhang et al., 2004;Liu et al., 2003;Liu et al., 2009;Liu et
al., 2011), to track the fate of $N_r$ in the present Bohai Sea. The observation data presented here are the basis for a combined mass and isotope balance model, results of which will be a basis for future studies on the rising impact of fast-growing mega-cities in the Bohai Sea catchment and their possible impact on the adjacent Yellow Sea.

## 2.    Materials and methods

### 2.1.    Sample collection

Research cruises were carried out by R/V *Dongfanghong 2* in spring and summer 2018 with 24 sampling sites in April and 25 sites in August, respectively (Fig. 1). Water samples were taken from several depths by 12 L Niskin bottles attached to a CTD rosette (911plus, Seabird, USA). The water samples were filtered using nucleopore polycarbonate filters (0.4μm) with plastic Nalgene filtration units. The filtered water was collected in Falcon PE tubes (45mL), frozen immediately (-20℃) and kept frozen until analyses in the home laboratory in Germany. Between 1 to 8 L of water were filtered through pre-weighted
GF/F filters (0.7μm, Φ=47mm, Sigma Aldrich) which had been pre-combusted at 450℃ for 4h. The filters were subsequently dried on board under 45℃ for 24 hours. Surface sediments were taken with a box corer and surface samples were transferred into plastic bags with a metal spoon, frozen at -20℃ and were kept frozen until later analysis in the home lab.

Yellow River water samples were taken from the Kaiyuan floating bridge in Lijin, located 44 km upstream of the river mouth. Samples were taken in the middle of the river course with a plastic reversing water sampler at 1 meter under the surface.
The water samples were filtered immediately for nutrient analysis and collection of suspended particles, and subsequently





were stored frozen until delivered to the home laboratory. Samples were taken monthly in May, July to November from Yellow River, and in November from Daliao River, Hai River, Luan River and Xiaoqing River (Fig. 1).

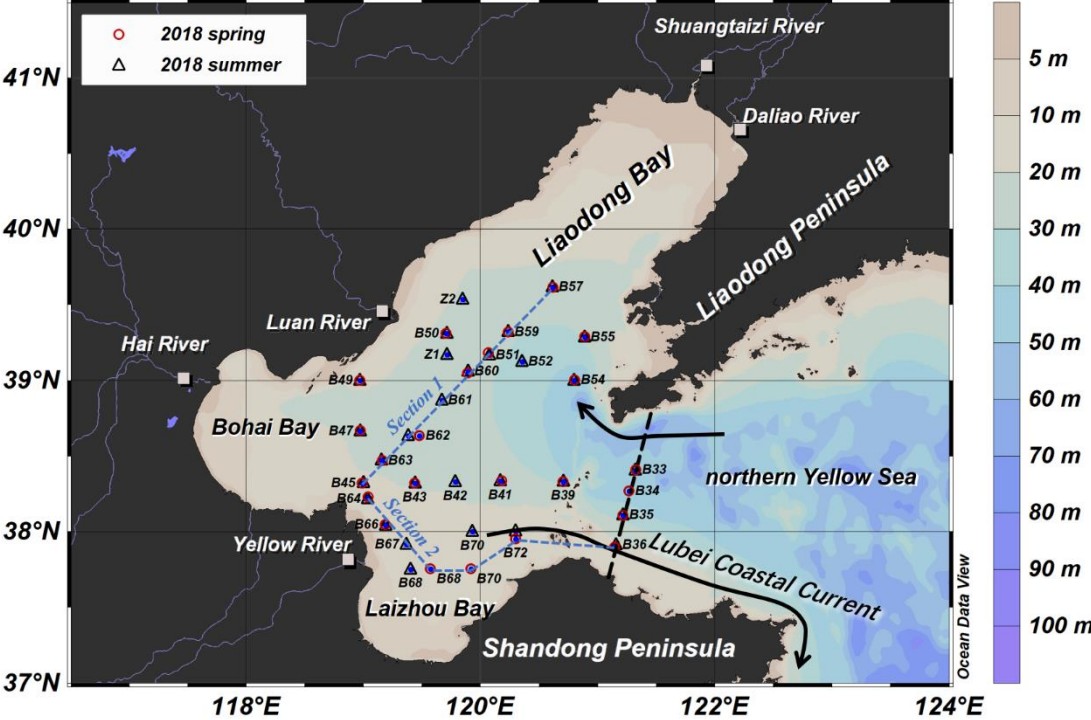

Fig. 1 Sampling sites in Bohai Sea and rivers (Xiaoqing River not shown). Open red circles stand for sampling site in
spring, open blue triangles stand for sampling sites in summer, names of the sites are marked nearby. Black arrows stand for the most significant currents flows in and out of the Bohai Sea. Blue dashed lines strand for two main sections, black dashed line stand for the boundary of our study area.

## 2.2.    Measurements of nutrients and nitrate isotopes

Nutrient concentrations were measured with an AutoAnalyzer 3 system (Seal Analytics) using standard colorimetric methods (Grasshoff et al., 2009). The relative error of duplicate sample measurements was below 1.5% for $NO_x$ and phosphate concentrations, below 0.3% for ammonium. The detection limit was < 0.5 μmol kg$^{-1}$ for $NO_x$, > 0.1 μmol kg$^{-1}$ for $PO_4^{3-}$, and >0.013 μmol L$^{-1}$ for ammonium.

$\delta^{15}N$ and $\delta^{18}O$ of nitrate ($\delta^{15}N=[(^{15}N/^{14}N)_{sample}/(^{15}N/^{14}N)_{standard}-1]\times1000‰$, $\delta^{18}O=[(^{18}O/^{16}O)_{sample}/(^{18}O/^{16}O)_{standard}-$
$1]\times1000‰$,) were determined with the denitrifier method (Sigman et al., 2001;Casciotti et al., 2002). Only the samples with nitrate concentrations >1.7μmol kg-1 were analyzed and $\delta^{15}N$ and $\delta^{18}O$ were analyzed in one sample run. Water samples were



injected into a suspension of the denitrifier *Pseudomonas aureofaciens* with injection volumes adjusted to yield 10 nmol $N_2O$. The $N_2O$ gas was purged by helium into a GasBench 2 (Thermo Finnigan) for purification. Afterwards the $N_2O$ gas was analyzed by a Delta V Advantage and a Delta V Plus mass spectrometer. Samples were measured in duplicate and the two

international standards IAEA-N3 ($\delta^{15}N\text{-}NO_3^-$=+4.7‰, $\delta^{18}O\text{-}NO_3^-$=+25.6‰) and USGS-34($\delta^{15}N\text{-}NO_3^-$=-1.8‰, $\delta^{18}O\text{-}NO_3^-$=-27.9‰) and an internal potassium nitrate standard were measured in each batch. The data were corrected by applying a bracketing correction (Sigman et al., 2009) and the standard deviations of the international and in-house standards was found to be ≤0.2‰ for $\delta^{15}N$ and ≤0.5‰ for $\delta^{18}O$. The standard deviations of duplicate samples were in the same range. Nitrite affects the results and was removed following the protocol of Granger and Sigman (2009) whenever $[NO_2^-]$ exceeded 5% of the $NO_x$

pool. In all other cases, we report combined (nitrite + nitrate) values.

### 2.3.    Measurements of suspended matters and sediments

The tared GF/F filters were weighed to calculate the amount of suspended particulate matter (SPM) per liter of water. Total carbon and nitrogen concentrations in SPM and sediment samples were measured by a Euro EA 3000 (Euro Vector SPA) Elemental Analyzer, and SPM samples with high carbon and nitrogen contents and sediments were acidified to measure organic

carbon content. The precision of total and organic carbon determination is 0.05%, that of nitrogen is 0.005%, and the standard deviations are less than 0.08 for total and organic carbon and 0.02 for nitrogen. Nitrogen isotope ratios were determined with a FlashEA 1112 coupled to a MAT 252 (Thermo Fisher Scientific) isotope ratio mass spectrometer. The precision of nitrogen isotope analyses is better than 0.2‰, and the standard deviation less than 0.03.

### 2.4.    Measurements of dissolved oxygen

The dissolved oxygen (DO) samples were collected, fixed, and titrated on board following the Winkler procedure at an uncertainty level of <0.5%. A small quantity of $NaN_3$ was added during subsample fixation to remove possible interferences from nitrite (Wong, 2012). The DO saturation (DO%) was calculated from field-measured DO concentration divided by the DO concentration at equilibrium with the atmosphere which was calculated from temperature, salinity and local air pressure, as per the Benson and Krause Jr (1984) equation.

### 130    2.5.    Hydrodynamic model of nutrient export from BHS to the Yellow Sea

The regional three-dimensional hydrodynamic Hamburg Shelf Ocean Model, HAMSOM (Backhaus, 1985), was applied in the East China Seas (23°-45°N, 117°-131°E) to calculate the water and nutrient transport through the Bohai Strait for the year 2018. The spatial resolution of the model is 2' (approx. 3.7 km) with 20 layers in vertical direction, while the calculation time step is 3 minutes. The topography data (resolution of 2') were obtained from marine navigation charts. The meteorological

forcing was derived from an hourly ERA5 dataset with a spatial resolution of 0.25° (CCCS, 2017). The open boundary SSH and the boundary T and S data and for the initial T and S fields were extracted from the daily Mercator-Ocean dataset (1/12





degrees) (Lellouche et al., 2019). 13 partial tides derived from the TPXO8-atlas v1 were superimposed to the SSH along the open boundary (Egbert and Erofeeva, 2002). The observed monthly river discharge were available for the two largest rivers, i.e., Changjiang and Yellow River (China, 2015-2018), while the inputs for the remaining rivers were derived from the Watergap dataset (0.5°, monthly climatology) (Müller Schmied et al., 2014).

Four sites on a north-south section through the Bohai Strait, i.e., B33, B34, B35, and B36, have been selected to represent the open boundary of the Bohai Sea (Fig. 1). The simulated SSH and current velocities (west-east-component) were extracted along this section. In addition, nutrient concentrations were interpolated from the observed data at the four sites to the grid of the hydrodynamic model along the Bohai Strait section. Since the observational data just include spring and summer values, the mean value of nitrate in spring and summer had to be extrapolated to an entire year.

## 3. Results

### 3.1. Hydrological properties and nutrients

#### 3.1.1. Hydrological properties

Averages of salinity and temperature in spring were 32.3±0.5 (n=72) and 4.7±0.8°C (n=72), respectively, and the water column was vertically mixed (Fig. 2 and Fig. 3). The YR discharged relatively warm water and the lowest salinity was observed in the southeast of YR estuary (site B68, T>6°C, S<31). Thus, the Yellow River Diluted Water (YRDW) is here defined as the water off the YR estuary with salinities lower than 31. In summer, averages of salinity and temperature were 31.6±0.8 (n=88) and 22.4±4.2°C (n=88), respectively, and the surface layer was stratified. The YRDW extended to an even larger area than in spring caused by high river discharge. The YRDW turned northeast towards LiaoDong Bay into the central BHS in the surface layer (T>27°C, S<31).

The water column oxygen concentrations (see Supplement 1) in the study area in spring and summer were 10.27-11.47 mg L and 3.84-8.86 mg L$^{-1}$, respectively, and thus much higher than the threshold for water column denitrification (0.15 mg/L). The detailed results of DO and other parameters are shown in Supplement 2.





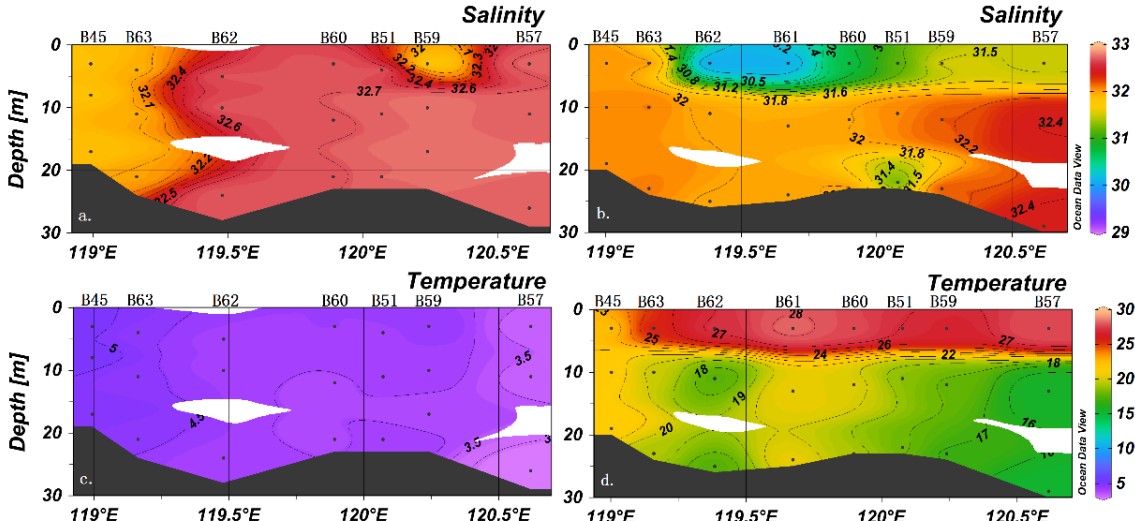

Fig. 2 Temperature (°C) and salinity (psu) of section 1 of spring (a. and c.) and summer (b. and d.).

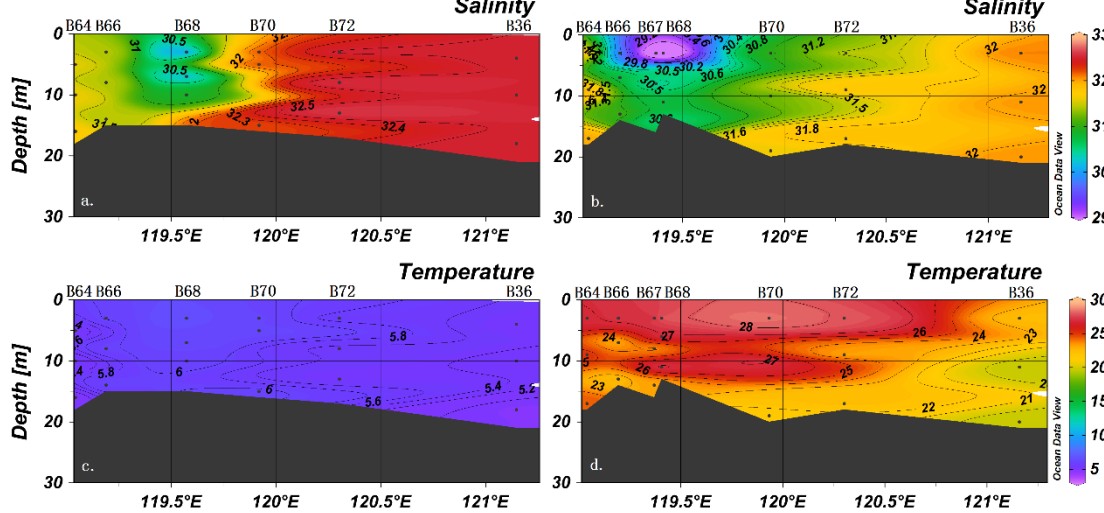

Fig. 3 Temperature (°C) and salinity (psu) of section 2 of spring (a. and c.) and summer (b. and d.).

**3.1.2.    Nutrient concentrations and distributions**

Nutrient concentrations in spring were vertically uniform, consistent with temperatures and salinities. The YR discharge is indicated by a nitrate maximum at station B68 (31.0 μmol/L) and the lowest observed salinity. Concentrations of $NH_4^+$ were





higher than 1 µmol/L and $PO_4^{3-}$ concentrations were lower than 0.4 µmol/L adjacent to YR estuary, and phosphate increased

towards the central BHS (along section 1) and north of Shandong peninsula (along section 2) (Fig. 4 and Fig. 5). The average

concentrations of $NO_3^-$, $NO_2^-$, $NH_4^+$ and $PO_4^{3-}$ of all spring samples were 6.5±5.8 µmol/L, 0.2±0.2 µmol/L, 0.8±0.5 µmol/L,

and 0.4±0.2 µmol/L (n=72), respectively.

In summer, nitrate was almost depleted in the BHS except for stations B66 and B67 (included in section 2) located in the

tongue of YRDW. $NO_2^-$ and $NH_4^+$, concentration was higher than 2 µmol/L and 5.5 µmol/L, respectively, in the same area

(Fig. 5). In the central BHS (section 1), the upper 10m water layer was nutrient depleted, while concentrations were high in

the lower layer. The average concentrations of $NO_3^-$, $NO_2^-$, $NH_4^+$ and $PO_4^{3-}$ of 1.9±2.7 µmol/L, 0.8±1.1µmol/L, 1.6±1.9

µmol/L and 0.1±0.1 µmol/L (n=85), respectively.

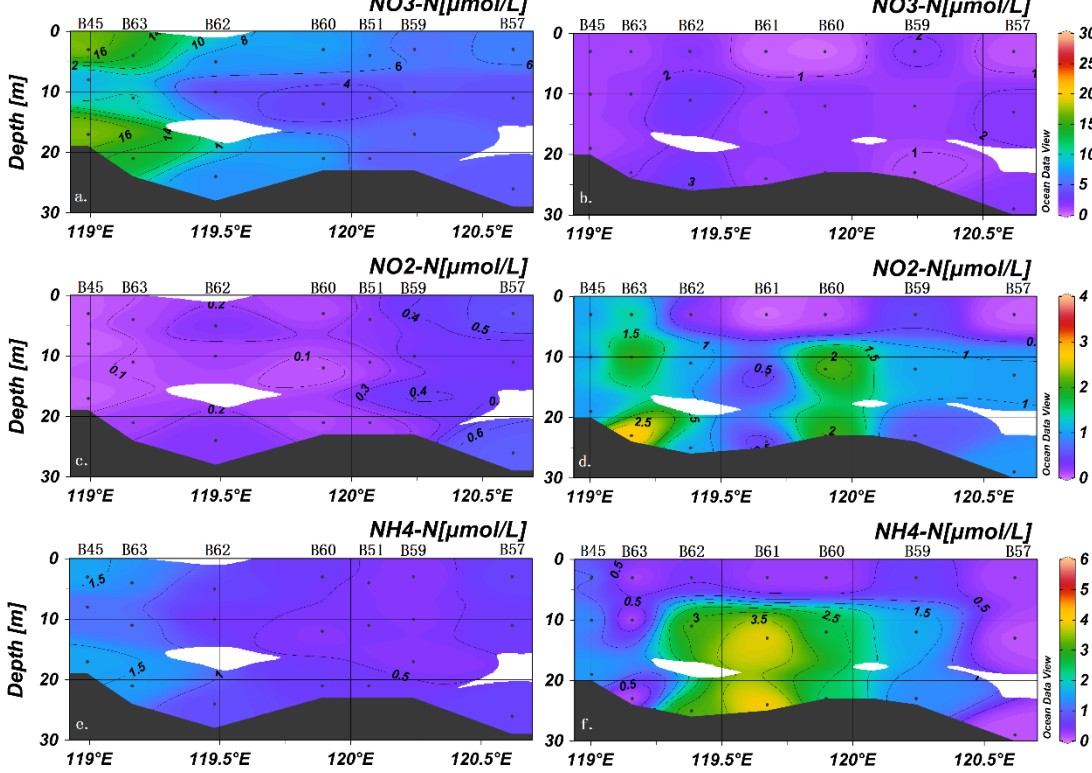

Fig. 4 Nutrients of section 1 (µmol/L) of spring (a., c., and e.) and summer (b., d., and f.).



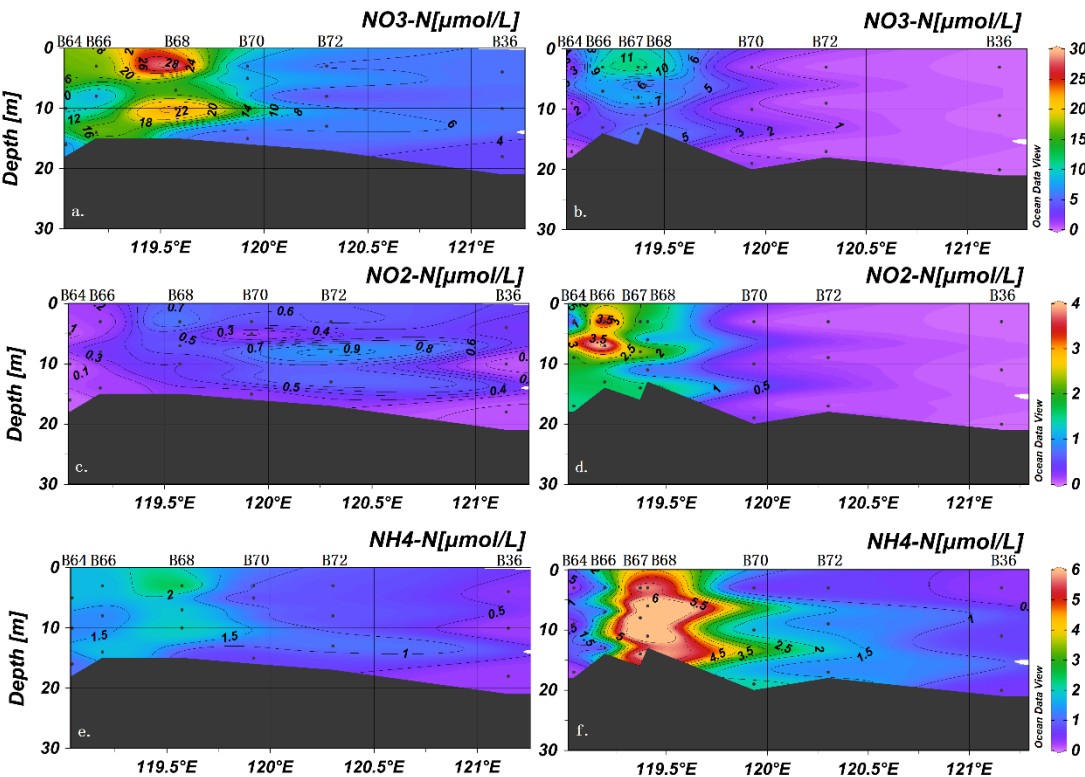

Fig. 5 Nutrients of section 2 (μmol/L) of spring (a., c., and e.) and summer (b., d., and f.).

## 3.2. Dual isotopes of nitrate

In spring, the average values of $\delta^{15}N$ and $\delta^{18}O$ were 7.8±1.4‰ and 12.0±3.3‰ (n=52), and $\delta^{15}N$ and $\delta^{18}O$ ranged between 5.6‰-10.5‰, and 5.0‰-20.0‰, respectively. $\delta^{15}N$ and $\delta^{18}O$ were both vertically homogenous, corresponding to the vertical profile of salinity, temperature, and nitrate. Relatively high $\delta^{15}N$ values were observed in the southwestern part of BHS including the areas adjacent to Bohai Bay and Laizhou Bay, which is consistent with high nitrate concentrations. The $\delta^{18}O$ was inversely related to $\delta^{15}N$ with low values near the YR estuary and high values in the northeast of the central BHS and north of Shandong peninsula (Fig. 6 and Fig. 7).

Due to the low nitrate concentration in summer, only a subset of samples could be analyzed and most of these are from the YRDW that had $[NO_3^-]$ >1.7μmol/L. The average values of $\delta^{15}N$ and $\delta^{18}O$ were 9.9±3.5‰ (n=23) and 8.7±3.3‰ (n=23), and ranged between 3.5‰-23.9‰ and 3.1‰-18.4‰, respectively. The mean value of $\delta^{15}N$ was higher than that of spring samples, whereas the $\delta^{18}O$ value was lower. Relatively high $\delta^{15}N$ and $\delta^{18}O$ values of 23.9‰ and 18.4‰ were registered in the surface water of YRDW (site B62) and decreased with water depths at increasing nitrate concentrations.



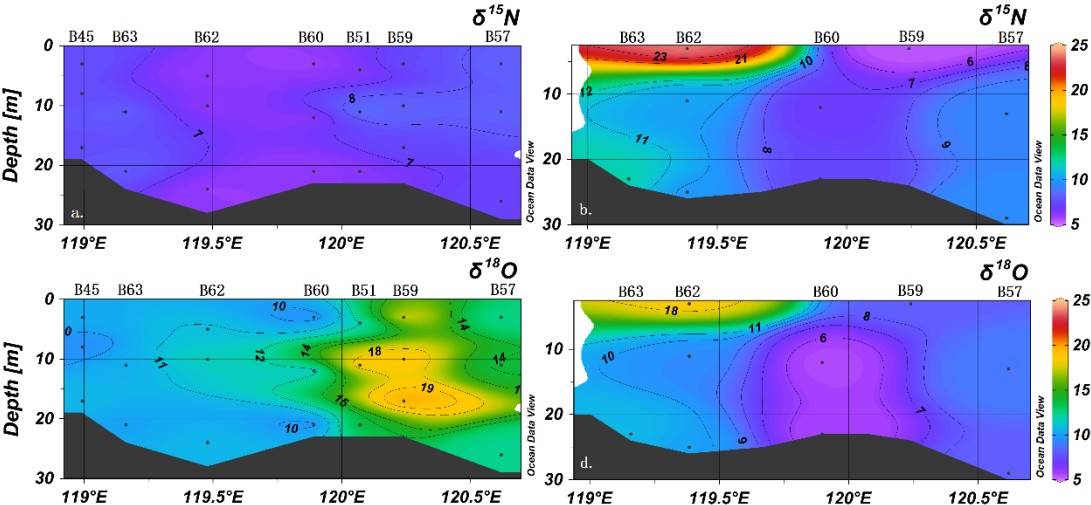

Fig. 6 Nitrate isotopes (‰) of section 1 of spring (a. and c.) and summer (b. and d.)

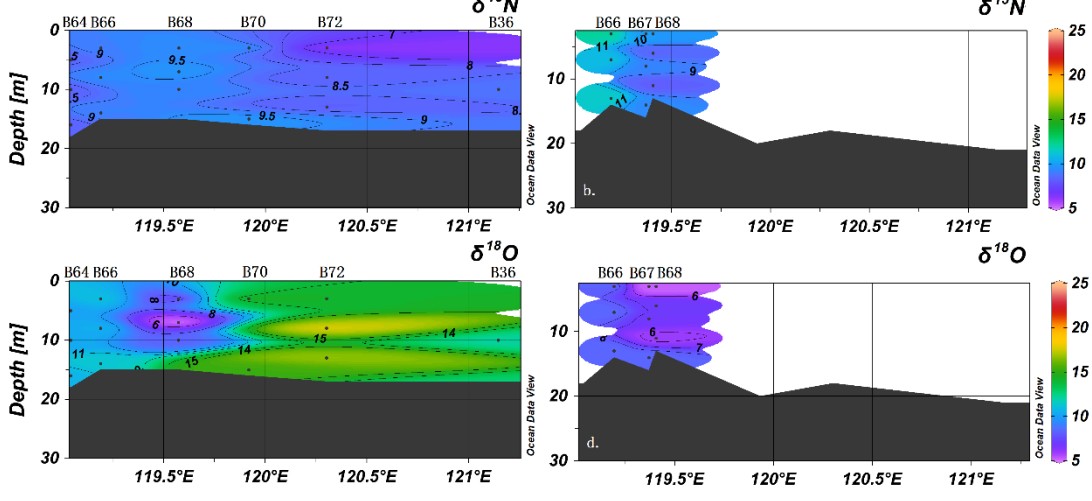

Fig. 7 Nitrate isotopes (‰) of section 2 of spring (a. and c.) and summer (b. and d.)

### 3.3. Suspended particulate matter

In spring suspended particulate matter (SPM) concentrations were mostly vertically homogenous along both transects with high values (>15mg L$^{-1}$) close to the YR mouth (see Supplement 3). $C_{org}$% and N% are anti-correlated with SPM concentrations and high values occurred in the central BHS and north of Shandong Peninsula. In summer SPM concentrations were





significantly higher than in spring and maxima occurred in deep water off the YR (>30 mg L$^{-1}$) and in the west part of the BHS. $C_{org}$% and N% maxima occur in surface waters in the eastern Laizhou Bay and the central BHS.

The average $\delta^{15}$N of SPM in spring was 4.8±0.9‰ (n=14), with a maximum (>6.10‰, n=3) south of Luan River. The lowest values were observed in the southern Bohai Strait and northeast of YR estuary. The other samples varied in a narrow range of 3.9‰-4.7‰ (n=11). In summer, the average of $\delta^{15}$N was 5.7±0.8‰ (n=34) and ranged from 3.9‰ to 7.2‰. Systematic variation of $\delta^{15}$N of SPM was barely discernable and only exhibited a weak decline from the YR mouth into the northeastern BHS (section 1) and into the BH Strait (section 2) (see Supplement 3), thus tracking the salinity dilution gradient in the surface 210 layer.

### 3.4. The discharge of the Yellow River

The water discharge of the year 2018 determined at the Lijin hydrography station was 333.8×10$^9$ m$^3$ which was by 14% higher than the multi-year average of 292.8×10$^9$ m$^3$ (1952-2015) (MWR, 2019). The monthly mean discharge was 27.80±20.21×10$^9$m$^3$month$^{-1}$ (n=12), which was higher than the multi-year average value by 14%-51%, indicating that in YR 215 basin 2018 was a flood year. The water discharge maximum was from July to October (Fig. 8). During May to November in 2018 (no value for June), the mass fluxes of nitrate, phosphate increased with the water discharge of YR, while the monthly fluxes of nitrite and ammonium exhibited an opposite tendency. $\delta^{15}$N and $\delta^{18}$O of YR nitrate ranged from 9.1‰ to 10.9‰ and 1.1‰ to 3.0‰, respectively. The monthly mass-weighted average value of $\delta^{15}$N and $\delta^{18}$O were 9.9‰ and 1.9‰, respectively. $\delta^{15}$N and $\delta^{18}$O were positively correlated ($\delta^{18}$O=1.04×$\delta^{15}$N-8.43, R$^2$=0.85), as were $\delta^{15}$N and the monthly mass flux of nitrate 220 (R$^2$=0.63).



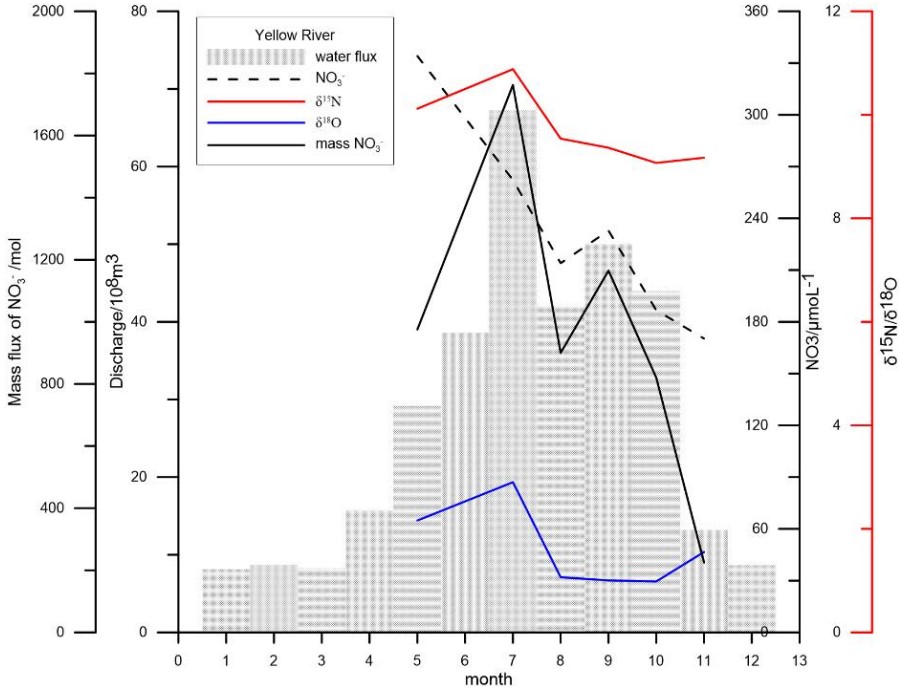

Fig. 8 Monthly variation of water flux (shadowed bar), nitrate concentration (dashed line), mass flux of nitrate (solid line), and dual nitrate isotopes ($\delta^{15}N$ in red solid line and $\delta^{18}O$ in blue solid line) of Yellow River in 2018. Water samples were monthly collected in May, July to November in Lijin, nitrate concentrations ($\mu mol\ L^{-1}$) and dual isotopes ($\delta^{15}N$ and $\delta^{18}O$, in per mil) were measured in home laboratory, monthly water flux data ($10^{8}\ m^{3}$)are according to the Yellow River Sediment Bulletin 2018 (http://www.yrcc.gov.cn/nishagonggao/2018/index.html#p=20 ).

### 3.5.  Nitrate exchange with the Yellow Sea

Based on current velocities and nutrient concentrations along the section crossing the BH Strait, the annual water and nutrient export from BHS to the Yellow Sea in the year 2018 was calculated to $1.26\times10^{-3}$ Sv (1 Sv=$10^{6}\ m^{3}/s$) and $0.9\times10^{9}$ mol $yr^{-1}$, respectively. In this study, the exported nitrate are assumed with the average isotopes values of the BHS ($\delta^{15}N=8.9‰$ and $\delta^{18}O=10.4‰$).

Making use of the three-dimensional model (HAMSOM) results, it is also possible to determine a spatial distribution of the annual nutrient flux through the Bohai Strait section (Fig. 9). Positive values represent a nutrient flux out of the Bohai Sea, while negative ones indicate a flux into the Bohai Sea. The strongest nutrient export occurs at the southern part of the Bohai Strait, while the major import takes place in the upper 15 m in the northern Bohai Strait (Fig. 9).

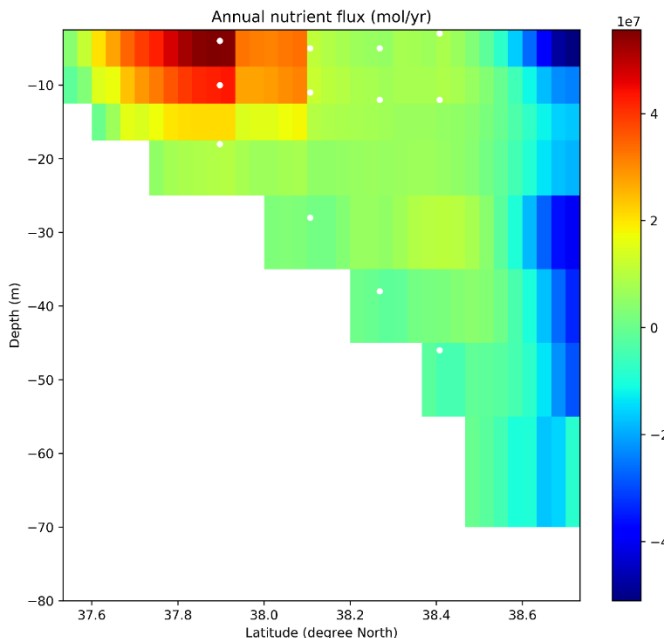

Fig 9 Simulated net fluxes of nitrate (mol yr$^{-1}$) in the Bohai Strait of the year 2018. White dots are the sampling sites, the color bar is the flux of nitrate ($10^7$ mol yr$^{-1}$), positive values stand for the export of nitrate from Bohai Sea to Yellow Sea.

## 4.  Discussion

### 4.1.  The hydrographic and nutrients characteristics in spring and summer

The sampling in early spring occurred during a season of low biological activity so that nutrients behaved almost conservatively. As is indicated by the negative correlations of $NO_3^-$ (r=0-.78, p<0.01), and $NH_4^+$ (r=-0.79, p<0.01) with salinity, the YR is one of the major sources of these nutrients in the BHS, whereas $PO_4^{3-}$ is contributed by the inflow of saline waters from the YS as indicated by the positive correlation with salinity (r=0.43, p<0.01). Concentrations of nitrate were relatively high in the southern Bohai Strait but low in its northern part, suggesting that in spring nitrate- rich water flows out of BH Strait along the northern shore of Shandong Peninsula in the LCC, while nitrate-depleted water flows in from the northern Yellow Sea via the northern strait (see the discussion of chapter 4.2.5).

In summer, the water is stratified with the thermocline (= halo- and nutricline) at about 8 m water depth. Nutrients are depleted to trace amounts above the thermocline. In contrast to the other nutrients, phosphate concentrations did not increase with depth in the southwestern part of BHS (i.e. Bohai Bay and Laizhou Bay). Similar to the spring situation, salinity was weakly positively correlated with $PO_4^{3-}$ (r=0.29, p<0.05) and $NO_2^-$ (r=0.32, p<0.05), whereas it was negatively correlated





with $NO_3^-$ (r=-0.69, p<0.01) and $NH_4^+$ (r=-0.37, p<0.01), respectively. The average N/P ratio in BHS in spring and summer was 28.2±38.2 and 86.9±126.3, respectively, implying that productivity in BHS was phosphorus limited. Thus, diazotrophic $N_2$ fixation is excluded as a significant input of $N_r$ because of high N:P ratios.

### 4.2.  The main sources and sinks of nitrate in the BHS

Most of the external and internal sources of nitrate to the BHS are characterized by distinct dual isotope values. These fingerprints combined with mass flux estimates are in the following used to constrain the mass and isotope budget of nitrate in the BHS. Specifically, the role of internal cycling processes can thus be quantified, which were lacking in previous budgets. In the following, each of the sources and sinks is described, along with isotope composition or isotope fractionation associated with cycling processes.

### 4.2.1.  Riverine inputs

The main input from this source of BHS $N_r$ is from the Yellow River and we calculated a nitrate input of YR of $7.95 \times 10^9$ mol yr$^{-1}$, based on the annual average discharge (Yellow River Conservancy Commission of MWR, 2019) and the load-weighted nitrate concentration during our sampling period during the year 2018. The water discharge of the other 7 important rivers (Hai River, Shuangtaizi River, Daliao River, Luan River, Xiaoqing River, Daling River and Xiaoling River) sums up to $59.86 \times 10^8$ m$^3$ yr$^{-1}$ (MWR, 2019;Ma et al., 2004;Yu et al., 2018;Zhang et al., 2004). Nitrate fluxes of these rivers were calculated based on water discharge and our nitrate measurements in YR, Hai River, Daliao River, Luan River and Xiaoqing River. Owing to a lack of data on nitrate concentrations of Shuangtaizi River, Daling River and Xiaoling River and considering that these rivers drain basins adjacent to Daliao River, we assume that nitrate concentration of these rivers were same as those of the Daliao River. The total riverine input of nitrate summed up to $9.49 \times 10^9$ mol.

The mass-weighted average annual values for $\delta^{15}$N and $\delta^{18}$O of nitrate in these rivers were 10.0‰ and 1.3‰, respectively, taken here to represent the river nitrate isotopic composition discharged into BHS.

### 4.2.2.  Submarine ground water input

The DIN supplied to BHS by submarine groundwater discharge (SGD) flux has been estimated to be 2-10 times the YR discharge (Luo and Jiao, 2016;Peterson et al., 2008;Wang et al., 2015). These fluxes of SGD are a mixture of submarine fresh groundwater discharge (SFGD) and recirculated saline groundwater discharge (RSGD) (Liu et al., 2011;Peterson et al., 2008), but only the freshwater component is relevant as a source for the budget.

The latest estimate of DIN flux for SFGD in Laizhou Bay amounts to 0.57-0.88 of YR input (Wang et al., 2015). We took the average of the SFGD flux (Wang et al., 2015) and the average nitrate concentration of ground water around Laizhou Bay





(there is no data on nutrient concentrations of SFGD available so far) (Zhang et al., 2016) to calculate the SFGD inorganic nitrogen flux, which resulted in $3.62\times10^9$ mol year$^{-1}$. The ratio of $NO_3^-/(NO_3^-+NH_4^+)$ is 0.75 in total SGD (Liu et al., 2011;Chen et al., 2007), so that the nitrate flux of SFGD is estimated at $2.73\times10^9$mol. Due to lack of data on SFGD from the entire BHS,

this number was used to represent the input from SFGD to the BHS, which may underestimate the factual value and is only approximately 10% of previous estimates of the input nitrate for SGD into BHS (Liu et al., 2011) due to the exclusion of RSGD.

Because of pollution and denitrification processes in soils, aquifers and groundwater (Zhang et al., 2013;Chen et al., 2007;Soares, 2000), the value of $\delta^{15}N$ and $\delta^{18}O$ of nitrate in SFGD is more enriched than those of river runoff and this is

290 illustrated by the observed $\delta^{15}N$ value of 20.2±9.0‰ (n=19) of on-land groundwater near the YR delta (Chen et al., 2007). As there are as yet no reported $\delta^{15}N$ values of SFGD and RSGD inputs, we decided to take this value as the signal of nitrate $\delta^{15}N$ imported by SFGD into the BHS. There are no data available for $\delta^{18}O$ of nitrate of SFGD, and we will discuss possible constraints in the box model discussion (Sect. 4.3.2).

### 4.2.3. Atmospheric deposition

Combined atmospheric input by wet and dry deposition ranged from $3.14\times10^9$ mol yr$^{-1}$ to $3.42\pm2.29\times10^9$ mol year$^{-1}$ (Zhang et al., 2004;Liu et al., 2003). We adopted the annual mass of $NO_x$ deposition for China of 6.2 Tg yr$^{-1}$ (Zhao et al., 2017) and related this value to the area of the BHS, which results in an annual deposition of $3.6\times10^9$ mol yr$^{-1}$. Owing to a lack of directly measured data for atmospheric $NO_x$ the BHS, we adopt $3.42\times10^9$ mol yr$^{-1}$ (Zhang et al., 2004) as the atmospheric nitrate flux.

The nitrate $\delta^{15}N$ values of PM2.5 (fine particulate matter suspended in the air) ranged from 3.5-17.8‰ in northern China (Fan et al., 2019;Zong et al., 2017;Song et al., 2020;Zhang et al., 2019), whereas nitrate $\delta^{15}N$ of precipitation ranged from -2.5 to +0.9‰ (Zhang et al., 2008;Chang et al., 2019;Li et al., 2019;Kim et al., 2019). Assuming that PM2.5 is the main component of dry deposition we use a $\delta^{15}N$ value of 8.2‰ reported from Beihuangcheng island in Bohai Strait (Zong et al., 2017), while the wet deposition has a $\delta^{15}N$ value of -2.35‰ (Chang et al. (2019). Wet deposition in BHS was estimated as 54%-68% of

total deposition (Liu et al., 2003;Zhang et al., 2008;Zhao et al., 2017), resulting in a mass-weighted average of BHS atmospheric deposition ranging from 1.03‰ to 2.56‰, of which the arithmetic mean value of 1.80‰ is adopted here.

The nitrate $\delta^{18}O$ of PM2.5 in BHS ranged from 65.0‰ to 88.1‰ (seasonally) (Zong et al., 2017), the value in Beijing is 88.3±6.9‰ (Song et al., 2020) and 57.80±4.23‰ in cloud samples of Shandong (Chang et al., 2019). For nitrate $\delta^{18}O$ of dry deposition, we thus assume a mean value of 80.5‰ and that wet deposition has a $\delta^{18}O$ of 57.8‰. These two estimates combined

give a ratio of dry and wet deposition in the range of 65.8‰ to 68.0‰. The arithmetic mean value is 66.9‰, which we take to represent the $\delta^{18}O$ of nitrate deposited from the atmosphere to the BHS.





The ammonium deposited from the atmosphere is assimilated by phytoplankton and is subsequently entrained into the N cycle via remineralization and nitrification or is nitrified directly in the water. Thus, the nitrified atmospheric ammonium is included here as a source bearing on $\delta^{15}N$ of nitrate in the sea water. The ammonium deposition in BHS was $6.15 \times 10^9$ mol yr$^{-1}$, which is more than the nitrate deposition of $3.42 \times 10^9$ mol yr$^{-1}$ (Zhang et al., 2004).

The atmospheric ammonium has low $\delta^{15}N$ values of -6.53 to -1.2‰ (Zhang et al., 2008;Chang et al., 2019). Given that the $\delta^{15}N$ value of ammonium of the north China Plain is -1.2‰ (Zhang et al., 2007), and that there is no obvious accumulation of ammonium in the surface layer in the observations, we assume that this isotope value is identical to the $\delta^{15}N$ value of nitrified atmospheric ammonium.

The $\delta^{18}O$ of nitrate from nitrification is roughly 1‰ higher than that of ambient $H_2O$ (Sigman et al., 2009;DiFiore et al., 2009;Casciotti et al., 2007;Casciotti et al., 2008b). The $\delta^{18}O$ of $H_2O$ in the BHS was reported as -0.67‰±0.25‰ (n=10) (Kang et al., 1994;Wu, 1991) and thus the $\delta^{18}O$ of the newly nitrified nitrate should be approximately 0.3‰.

### 4.2.4. Benthic fluxes

#### 4.2.4.1. Nitrate diffusing from water to sediment

Benthic fluxes of nutrients have been investigated through incubation experiments and diffusion models (Zhang et al., 2004;Liu et al., 2011) and range from 63.3±296×10$^6$ mol/month nitrate diffusing from bottom water into the sediments (Liu et al., 2011) to sediment-water effluxes of $4.28 \times 10^9$ mol year$^{-1}$ in the box model of Zhang et al. (2004). The difference of the estimates is probably due to different methods of calculation. For this study we adopted the annual flux from water to sediment as $0.76 \times 10^9$ mol year$^{-1}$ (Liu et al., 2011). We assume that diffusion is not accompanied by isotope fractionation, so $\delta^{15}N$ and $\delta^{18}O$ of nitrate diffusing into the sediment was assumed to be the same as the nitrate pool in BHS (8.9‰ and 10.4‰, respectively).

#### 4.2.4.2. Ammonium diffusing from sediment to water

The processes of nitrogen cycling in sediments are complex and variable (Lehmann et al., 2004). The degradation of organic matter, nitrification and assimilation are acting under aerobic conditions, whereas denitrification, anammox and dissimilatory nitrate reduction to ammonium (DNRA) are observed under anaerobic conditions. When organic matter is degraded in the surface sediments, part of the produced ammonium diffuses into the overlying bottom water and subsequently is nitrified to nitrite and nitrate under aerobic condition. For our purpose only the ammonium nitrified bears on the seawater nitrate pool. The mean $\delta^{15}N$ value of sediment in BHS was 5.4‰ (n=20), and according to the fractionation factor during organic matter remineralization of 2‰ (Möbius, 2013) and subsequent nitrification (see above), the $\delta^{15}N$ and $\delta^{18}O$ of nitrate efflux from sediment are assumed to be 3.4‰ and 0.3‰, respectively.





### 4.2.5. Sedimentation

The mass flux of $N_r$ sedimentation is unknown. In terms of the effects of $N_r$ sedimentation on nitrate dual isotopes, phytoplankton organisms that assimilate nitrate from the dissolved phase are the main source of sinking particles, so that the N and O will be removed from the nitrate pool following the assimilation fractionation factor. Sinking particles in the BHS have a $\delta^{15}N$ of 5.2‰ ($\delta^{15}N_{sink}$), which integrates multiple processes such as photosynthesis of phytoplankton, heterotrophic synthesis of bacteria, and heterotrophic degradation (remineralization).

There is no observed data of $\delta^{18}O$ of nitrate removed from the pool during assimilation ($\delta^{18}O_{sink}$), but this value can be estimate by the assimilation fractionation factor ($^{18}\varepsilon$). The per mil fractionation factors $\varepsilon$ of N ($^{15}\varepsilon$) and O ($^{18}\varepsilon$) in nitrate during assimilation are generally assumed to be around 5‰, so that $^{15}\varepsilon$:$^{18}\varepsilon$=1:1. Here we adopt the average of $^{15}\varepsilon$ and $^{18}\varepsilon$ as 5‰ (Granger et al., 2010;DiFiore et al., 2009;Liu et al., 2017;Wu et al., 2019;Umezawa et al., 2013;Wang et al., 2016), so that the $\delta^{18}O$ of nitrate removed from the pool during assimilation ($\delta^{18}O_{sink}$) should be 5.0‰ according to the $\delta^{18}O$ (10.0‰) of the dissolved nitrate pool.

## 4.3. The nitrate budget in the BHS

A box model of the nitrate budgets for the Bohai following the LOICZ approach (Zhang et al., 2004) balanced sources and sinks of nitrate in BHS and was updated by several other nitrate budgets for the BHS during last two decades (Zhang et al., 2004;Liu et al., 2003;Liu et al., 2011;Liu et al., 2009). All were, in general, not completely constrained because of a lack of data on some important source or loss terms. We here associate the nitrate isotope compositions of pools, sources, and sinks of nitrogen with a box model of the BHS nitrate in order to improve the understanding of nitrate cycling in the BHS. Finally, based on the combined mass and isotope box model informed by new data on the isotopic composition of nitrate, surface sediment, and suspended particulate nitrogen in the water column discussed above, we propose an updated N-budget that is internally consistent.

### 4.3.1. The nitrate budget based on mass fluxes and corresponding $\delta^{15}N$ values

In our hypothesis, the sources of nitrate for BHS are river inputs, submarine fresh ground water input, atmospheric deposition, and remineralization. Most important sinks are net export to the YS, sediment denitrification and particulate matter sedimentation. Assuming the mass and N isotope of nitrate in the BHS are in steady state, the sources and sinks of nitrate follow the Eq. (1) and Eq. (2):

$$(m_{atm} + m_r + m_N + m_{SFGD} + m_{ntr}) - (m_{net} + m_{sink} + m_{denitr}) = 0 \tag{1}$$

$$\left(\delta^{15}N_{atm}m_{atm} + \delta^{15}N_r m_r + \delta^{15}N_N m_N + \delta^{15}N_{SFGD}m_{SFGD} + \delta^{15}N_{ntr}m_{ntr}\right) - \left(\delta^{15}N_{net}m_{net} + \delta^{15}N_{sink}m_{sink} + \delta^{15}N_{denitr}m_{denitr}\right) = 0 \tag{2}$$





where the terms m with different subscripts refer to the corresponding nitrogen mass fluxes, $m_{atm}$ refers to atmospherically

deposited nitrate,, $m_r$ refers to river nitrate, $m_N$ refers to nitrified ammonium deposited from the atmosphere, $m_{SFGD}$

refers to nitrate in submarine fresh groundwater discharge, $m_{ntr}$ refers to nitrification in the water column. In terms of sinks,

$m_{net}$ refers to the mass fluxes associated with net export of nitrate from BHS to the YS, $m_{sink}$ refers to nitrate sedimenting

from seawater as particulate N, and $m_{denitr}$ refers to denitrification in the sediment. The unit of the mass fluxes is $10^9$mol.

The "$\delta^{15}$N" refers to the $\delta^{15}$N value of the N mass flux which with the same subscripts. As mentioned previously, the mass

fluxes for $m_N$, $m_{ntr}$, $m_{sink}$ and $\delta^{15}N_{ntr}$ are unknown and need to be constrained.

       The range of $\delta^{15}N_{ntr}$ can be constrained by a simplified interior nitrate cycling model. Ammonium links particles and

nitrate in this interior cycling, and there are two different sources of remineralised ammonium in seawater. One is ammonium

diffusing from the sediment, the other is the ammonification of PN in the water column. The ammonium from remineralisation

through both processes is then nitrified in the water column and is a source of nitrate. The average $\delta^{15}$N values of PN and

sediments in the BHS in our study are 5.2‰ and 5.4‰, respectively. The fractionation factor of ammonification of PN and

sediment as the first step of generating recycled nitrate are estimated to 3‰ (Sigman and Fripiat, 2019) and 2‰ (Möbius,

2013), respectively. The remineralised ammonium from PN and sediments thus should have $\delta^{15}$N values between 2.2‰ and

3.4‰.

The ammonium concentrations in the BHS are low in the water column in spring and in the surface layer in summer,

indicating that the ammonium from PN mineralization is most likely completely converted to nitrate, so that there is no

fractionation effect for this step. Thus, the $\delta^{15}$N value of newly nitrified nitrate from complete nitrification of ammonium

generated by PN mineralization is 2.2‰. In the case of incomplete nitrification, especially under the thermocline in summer,

the newly nitrified nitrate has a $\delta^{15}$N of 0.2‰, given a net fractionation factor of nitrification (ammonium to nitrate) of 2‰

(Sigman and Fripiat, 2019). In our model below, the assimilation of ammonium originating from SPM remineralization was

not included, as its proportion is unknown in the BHS. The simplified model may thus underestimate the input of $^{15}$N-depleted

nitrogen into the nitrate pool.

       Ammonium diffusing out of the sediment will either be mixed into the euphotic layer and subsequently assimilated by the

phytoplankton, or nitrified in the water column. The accumulating of ammonium beneath the thermocline was a significant

process in summer, as shown by the high nitrite and ammonium concentrations beneath the thermocline. The ratio of these two

branching processes is not known in the BHS. If all of the ammonium from sediment is nitrified, the produced nitrate will have

a $\delta^{15}$N of 3.4‰ (see above). If the ammonium is only partially nitrified (especially in summer beneath the thermocline), the

produced nitrate will have a $\delta^{15}$N of 1.4‰ at a fractionation factor of nitrification of 2‰ (Sigman and Fripiat, 2019). Thus, the

$\delta^{15}$N value of the nitrate produced by nitrification ($\delta^{15}N_{ntr}$) of ammonium from sediment is in the range of 1.4‰ to 3.4‰.





Overall, combing the $\delta^{15}$N ranges of PN- and sediment- originated nitrate, the range of $\delta^{15}$N for newly nitrified nitrate is 0.2‰ to 3.4‰.

### 4.3.2.    The coupled N and O budgets box model of nitrate

Because the nitrate mass fluxes $m_N$, $m_{ntr}$, and $m_{sink}$, cannot be segregated only based on N mass budgets and $\delta^{15}$N values, we turn to the $\delta^{18}$O values of the sources and sinks of nitrate for further constraints. Eq. (3) applies if we assume that

the oxygen isotope composition of nitrate reflects the steady-state mass fluxes, as does $\delta^{15}$N of the nitrate pool:

$$\left(\delta^{18}O_{atm}m_{atm} + \delta^{18}O_r m_r + \delta^{18}O_{ntr}m_N + \delta^{18}O_{SFGD}m_{SFGD} + \delta^{18}O_{ntr}m_{ntr}\right) - \left(\delta^{18}O_{net}m_{net} + \delta^{18}O_{sink}m_{sink} + \right.$$
$$\left. \delta^{18}O_{denitr}m_{denitr}\right) = 0 \tag{3}$$

where the $\delta^{18}$O subscripts refer to the nitrate mass flux with the same subscripts. The $\delta^{18}$O of different sources and sinks are either fixed values or ranges of values in our own data, or those taken from the literature (Table 1).

According to Eq. (1), (2) and (3), the unknown mass fluxes $m_N$, $m_{ntr}$ and $m_{sink}$ can be solved by a set of ternary linear equations including the three unknown terms, when appropriate boundary values of $\delta^{15}$N$_{ntr}$ and $\delta^{18}$O$_{SFGD}$ are chosen. $\delta^{15}$N$_{ntr}$ ranged in 0.2‰ to 3.4‰ as discussed in Sect. 4.3.1, whereas $\delta^{18}$O$_{SFGD}$ is a crucial term without any data or literature constraint. As the only constraint, $\delta^{18}$O$_{SFGD}$ is expected to be higher than the value of nitrate imported from the rivers ($\delta^{18}$O$_r$=1.3‰) due to the fractionation associated with denitrification in the anaerobic aquifers (see 4.2.2). The results can be summed up in three

different cases:

(1)When setting the value of $\delta^{15}$N$_{ntr}$ to 3.4‰, we obtain estimates for $m_N$ that range from 0.00 to $1.31\times10^9$mol year$^{-1}$, for $m_{ntr}$ in the range of 32.57 to $38.69\times10^9$mol year$^{-1}$ and $m_{sink}$ in the range of 47.87 to $52.68\times10^9$mol year$^{-1}$. The corresponding values of $\delta^{18}$O$_{SFGD}$ range from 1.3‰ to 16.3‰, and the upper range of $\delta^{18}$O$_{SFGD}$ yields $m_N$=0.0 due to the assumption that any mass flux must be equal or greater than 0.

(2) When we choose a $\delta^{15}$N$_{ntr}$ value of 3.2‰ and 3.0‰, respectively, to explore effects of the methodological error of $\delta^{15}$N for our isotope method (0.2‰, see Sect. 2.2), again under the premise that the mass fluxes are positive numbers, results in $m_N$, $m_{ntr}$ and $m_{sink}$ estimates in a narrower range than when $\delta^{15}$N$_{ntr}$ is 3.4‰; these results are not shown.

(3) $\delta^{15}$N$_{ntr}$ values lower than 3.0‰ result in $\delta^{18}$O$_{SFGD}$ lower than 1.3‰, which is highly unlikely.

Thus, reasonable solutions only are reached when $\delta^{15}$N$_{ntr}$ is between 3.4‰ and 3.0‰. The results of the budget are shown

in Fig. 10 and Table 1, respectively.



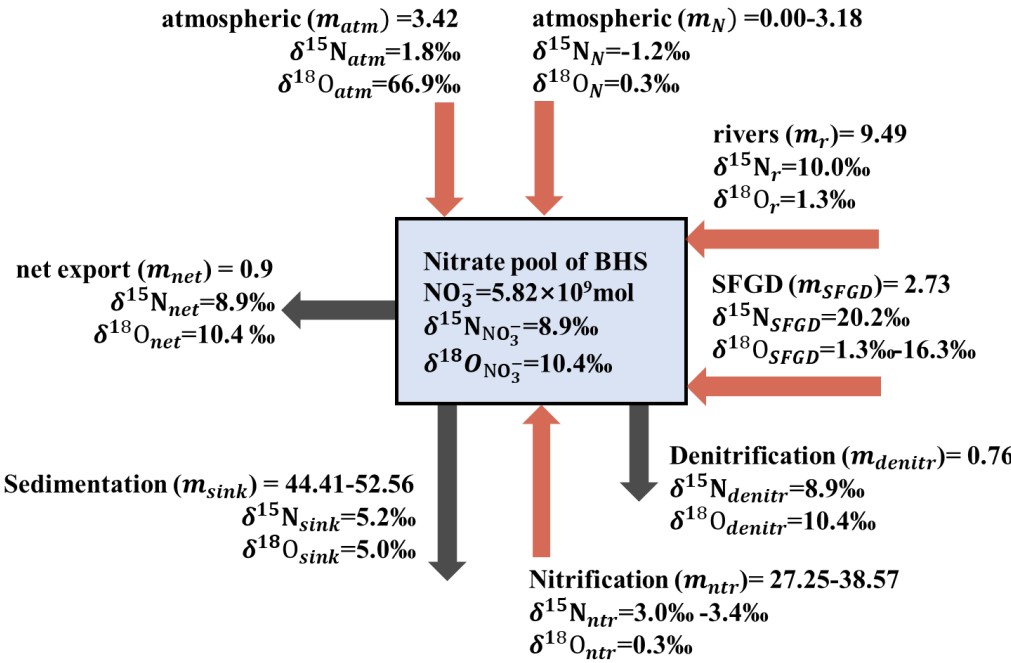

Fig. 10 The budgets and the corresponding dual isotope values of nitrate in the BHS 2018. The terms included are discussed in the text in section 4.3. The unit of mass flux is $10^9$ mol yr$^{-1}$.



Table 1 Sources and sinks and the corresponding δ15N and δ18O values of nitrate in the BHS

| | Contribution ions | Mass fluxes 10⁹mol $NO_3^-$ | References | δ15N | References | δ18O | References |
|---|---|---|---|---|---|---|---|
| **Sources** | | | | | | | |
| Atmosphere (nitrate) | 6.3%-6.9% | 3.42 | Zhang et al. (2004) | 1.8‰ | Assumption based on Zong et al. (2017);Chang et al. (2019);Liu et al. (2003);Zhang et al. (2008);Zhao et al. (2017), see section 4.2.3 | 66.9‰ | Assumption based on Zong et al. (2017);Song et al. (2020);Chang et al. (2019);Liu et al. (2003);Zhang et al. (2008);Zhao et al. (2017), see section 4.2.3 |
| Atmosphere (nitrified ammonium) | 0.0-2.6% | 0.00-1.31 | Assumption based on this study, see section 4.2.3 | -1.2‰ | Assumption based on Zhang et al. (2007);Sigman et al. (2005), see section 4.2.3 | 0.3‰ | Assumption based on Casciotti et al. (2008a);Sigman et al. (2009);DiFiore et al. (2009);Casciotti et al. (2007);Kang et al. (1994);Wu (1991), see section 4.2.3 |
| Rivers | 17.8%-19.8% | 9.49 | Assumption based on Ma et al. (2004);Yu et al. | 10.0‰ | This study | 1.3‰ | This study |





| Process | (%) | value | Reference | (‰) | value | Reference | (‰) | Reference |
|---|---|---|---|---|---|---|---|---|
| SFGD | 5.0%-5.5% | 2.73 | (2018);Zhang et al. (2004) and this study, see section 4.2.1 | 20.2‰ | 9.5 | Chen et al. (2007) | 1.3‰- | Assumption based on this study, see 4.2.2 and 4.3.2 |
| Nitrification | 65.8%-71.2% | 32.57-38.69 | Assumption based on Wang et al. (2015);Liu et al. (2011);Chen et al. (2007), see section 4.2.2 | 3.2‰-3.4‰ | | Assumption based on Sigman and Fripiat (2019);Möbius (2013), see section 4.3.2 | 0.3‰ | Assumption based on Casciotti et al. (2008a);Sigman et al. (2009);DiFiore et al. (2009);Casciotti et al. (2007);Kang et al. (1994);Wu (1991), see section 4.2.3 |
| **Sinks** | | | | | | | | |
| Net export | 1.7%-1.8% | 0.9 | Assumption based on this study, see section 3.5 | 8.9‰ | | Assumption based on this study, see section 3.5 | 10.4‰ | Assumption based on this study, see section 3.5 |
| Sedimentation | 96.7%-97.0% | 47.87-52.68 | Assumption based on this study, see section 4.3.2 | 5.2‰ | | Assumption based on this study, see section 4.2.5 | 5.0‰ | Assumption based on this study, see section 4.2.5 |






| Denitrification | 1.4%-1.5% | 0.76 | Liu et al. (2011) | 8.9‰ | Assumption based on this study, see section 4.2.4 | 10.4‰ | Assumption based on this study, see section 4.2.4 |



The mass flux of nitrate ($m_N$) originating from nitrification of atmospheric ammonium ranges from 0.00 to $3.18 \times 10^9$ mol year$^{-1}$ and accounts for up to 51.7% of the total ammonium deposition ($6.15 \times 10^9$ mol year$^{-1}$; Zhang et al. (2004). This in turn implies that most of the atmospherically deposited ammonium is directly assimilated rather than nitrified to nitrate. This agrees with phytoplankton preference to assimilate ammonium rather than nitrate (Glibert et al., 2016). The bulk of internal sources of nitrate originates from nitrification in the water column (from water column ammonification and ammonium diffusing from sediment). This single source ($m_{ntr}$) accounts for 59.1%-71.2% of the total sources of nitrate in BHS and appears to be much more important than in other coastal environments. For example, between 15%-27% of productivity was supported by nitrified ammonium in the seawater in Monterey Bay (Wankel et al., 2007). Likewise, nitrification supplied 34% of the surface nitrate in the eastern Hainan Island, which like Monterey Bay is also an upwelling area (Chen et al., 2020). This indicates that nitrate regeneration by nitrification may play a more important role in shallow and land-input dominated marginal seas than in upwelling dominated marine settings.

### 4.3.3.  Assessment of model uncertainties

The uncertainties of the modeled nitrate mass and isotope budget lie essentially in some of the mass flux estimates and in possibly erroneous assumptions on values of $\delta^{15}$N and $\delta^{18}$O in some of the sources and sinks. The mass flux of atmospheric deposition of nitrate is reliable, because differences in estimates from previous studies are quite small. There may be small deviations of $\delta^{15}$N$_{atm}$ and $\delta^{18}$O$_{atm}$ due to the combination of data for wet and dry deposition sampled separately by different methods, The largest uncertainties in the nitrate budget are associated with SFGD, in terms of both the mass flux ($m_{SFGD}$) and the nitrate isotopes ($\delta^{15}$N$_{SFGD}$ and $\delta^{18}$O$_{SFGD}$), for which better estimates are needed. Furthermore, underestimation of sources of nitrate with low $\delta^{15}$N and $\delta^{18}$O values cannot be excluded, such as the share of remineralised ammonium (from sinking particles and diffusing out of sediment) assimilated directly by phytoplankton.

### 4.3.4.  Biogeochemical implications of the box model

In other coastal eutrophic regions, such as the North Sea, a high $\delta^{15}$N of river nitrate is reflected in a halo of high $\delta^{15}$N in surface sediments in offshore areas (Pätsch et al., 2010). In the Bohai Sea, such an isotopic halo of river-borne eutrophication is not observed despite similar water exchange rates of 1-2 years (Li et al., 2015;Serna et al., 2010) and similarly isotopically enriched river inputs. We speculate that the lack of a fingerprint of river nitrate in the d$^{15}$N of sediments of BHS may be masked by active nitrification and atmospheric deposition that rapidly eradicate and homogenise spatial gradients.

Despite the uncertainties that are related to the box model approach, combining mass and isotope budgets of nitrate sources and sinks is clearly superior to solely nitrate mass balance considerations, especially when it comes to segregating the anthropogenic nitrate and the recycled nitrate inputs. It is of note that the Bohai Sea does not appear to pass on significant



amounts of nitrate to the adjacent northern Yellow Sea, so that the effects of excessive loading of this shallow mixing zone between land and ocean with anthropogenic nitrogen are yet mitigated by internal cycling processes.

## 5 Conclusions

Rivers contributed 17.5%-20.6% and the combined terrestrial runoff (including submarine discharge of nitrate with fresh ground water) account for 22.6%-26.5% of the total $N_r$ input to the BHS. Atmospheric input contributes 6.3%-7.4% of nitrate to the BHS. Nitrification contributes 59.1%-71.2% of the total nitrate, indicating an unusually active interior nitrogen cycling of in the BHS. Nitrate was mainly trapped in the BHS and only very little was exported to the YS (only 1.7%-2.0%). Furthermore, nitrate was rather assimilated than exported to the YS along the main transport pathway Lubei Coastal Current, effectively retaining $N_r$ in BHS. Sedimentation trapped 96.4%-96.9% of nitrate inputs, whereas denitrification was only active in the sediments that removed 1.4%-1.7% of nitrate from the pool. Seasonal biogeochemical variations were observed in the BHS in that dissolved inorganic nitrogen increased during summer under the thermocline, implying significant biological regeneration. If the interior cycling increases, for instance fueled by increased terrestrial and atmospheric $N_r$ inputs, respiration coupled to organic matter and N recycling will increase and water-column hypoxia could consequently spread in the future and compromise ecosystems in the BHS. Whether this will invigorate water-column denitrification to balance the additional inputs is an open question, as is the capacity of BHS as a nitrate buffer between the growing source of $N_r$ on land and the open ocean.

## Methods Appendix:

Nitrate was reduced to nitrite with a copperized cadmium column first. The nitrite ions reacted with sulfanilamide and N-1-naphthylethylendediamine (NEDD) to form red azo dye, and then measured at 520-560nm. Phosphate determination followed the method of Murphy and Riley (Murphy and Riley, 1962). Under acid conditions a phosphomolybdic complex was formed of ortho- phosphate, antimony and molybdate ions (Wurl, 2009). Followed by the reduction of ascorbic acid, the blue colour complex was measured at 880 nm. The sample with ammonium is reacted with o-phthalaldehyde (OPA) at 75°C in the presence of borate buffer and sodium sulfite to form a fluorescent species proportional to the ammonia concentration. The fluorescence is measured at 460 nm following excitation at 370 nm(Kérouel and Aminot, 1997). Silicate is reacted with ammonium molybdate to silicomolybdate, and reduced in acidic solution to molybdenum blue by ascorbic acid (Grasshoff et al., 2009).

## Data availability

Data are prepared to be uploaded to the Pangaea database (https://doi.pangaea.de/ ).





**The contribution description:**

BG and KCE designed the study. ST collected the samples on board. JT and YL supported the sampling on cruises and field trips. ST, NL, and TS analyzed samples. TX and WZ supplied the dissolved oxygen data. WL ran the HAMSOM model and calculated water mass exchanges. ST, BG, KCE and KD interpreted the data. ST prepared the manuscript with input from all co-authors.

**Competing interests**

The authors declare that they have no conflict of interest

**Acknowledgment:**

The study was part of the project FINGBOYEL (03F07686B) carried out within the MEGAPOL consortium and financed by the German Federal Ministry of Education and Research (BMBF). We thank the crew of the Chinese research vessel *Dongfanghong 2* for their support of our work on board. We are indebted to Yuan Li from Yantai Institute of Coastal Zone
Research for sampling the Yellow River. We thank Markus Ankele and Frauke Langenberg for their support of the nutrient and stable isotope analyses.

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
