# Peer review of "A nitrate budget of the Bohai Sea based on an isotope mass balance model"

_Biogeosciences, 2020_

## Referee Comment (RC1) · Anonymous Referee #1 · 16 Mar 2021

The manuscript of Tian et al. presents a budget of nitrogen from the Bohai Sea in China and also measurements along two transects in the bay which are impacted by large river inflows. Samples for nitrate and POM analyses had been collected during spring and summer seasons and stable isotopes been measured. Fluxes in and out of the bay were modelled. Moreover, a budget using the LOICZ approach was constructed using measurements but mainly data from other studies. The budget balances sources and sinks of nitrate and is supported by stable isotope data of nitrate. As a central conclusion the overwhelming role of nitrification as the major source of nitrate is presented. Overall the topic is of interest and fits well to the scope of the journal Biogeosciences.

Unfortunately, there are several major problems with this manuscript affecting the combination of measurements with the budget, the equations, and the conclusion. The most significant error is the definition of equations (1) and (2). Sources and sinks are listed and supposed to be balanced. However, the source terms list nitrification and the loss terms list sedimentation. Both of these do not fulfill the criteria of a source or sink, respectively.

Nitrification is neither a source nor a sink for nitrate but simply a microbial process that converts ammonium via nitrite to nitrate. Nitrification does not generate new DIN for a system simply because the substrate of the nitrification process is ammonium and comes from internal turnover processes of organic matter. The LOICZ report (no 5 LOICZ BIOGEOCHEMICAL Modelling Guidelines, 1996) states "The important point to note with this reaction (nitrification) is that carbon and phosphorus are not directly involved in the net reaction. Again this makes the point that the relationship between NO3 and NH4 may be considered an "internal cycle" which need not be dealt with directly."

Sedimentation is also a problematic variable for a balanced budget because it is not a removal process for nitrate. Organic material in sediments is prone to remineralization and resuspension. Only if that organic matter is permanently buried it can be considered a long-term sink. The finding that nitrification in the Bohai Bay is a very active process – generating large quantities of nitrate - is not a surprise but rather typical for coastal eutrophied systems.

Another problem I have with this manuscript is that it actually consists of two independent stories; one is the field study of nitrogen compounds and stable isotopes in the Bohai Sea and the other the budget. Both are rather disconnected although the authors try to include some measurements in the budget. But data from only two seasons can hardly be used for a budget averaging annual mean fluxes. The field data are rather distracting from the main scope because they suggest that part of the budget is based on measurements although, most data are taken from other publications. Of course

the authors write very clearly where the numbers for the budget are from and which underlying assumptions were applied to derive mean values. Nevertheless, the field data and budget remain two different stories. The results of the HAMSON model were not used at all and in the discussion the field data are only briefly mentioned.

The final major concern is the lacking error estimate of the budget. All field data are subject to some degree of major or minor inaccuracy, which is not analyzed and not included in the budget calculations. Point 4.3.3 is insufficient and only addresses single sources. What's needed is an error propagation estimate.

In summary, it is very sad to state that this study is neither conclusive and correctly done nor represents scientific progress – although a lot of effort has been invested by the authors. May be the field data can be presented elsewhere and the HAMSON model data used to constrain the "metabolism" of the bay in spring and summer. The budget may be corrected amended with an error estimated and submitted elsewhere. A clear research question then needs to be defined.

In case the authors decide to revise and refocus the manuscript, there are a smaller issues that would need consideration:

The introduction would benefit from a clearer focus and research hypothesis instead of an aim. In its present form the paragraphs read a bit isolated from each other.

Line 33 Reactive nitrogen is different to fixed nitrogen. While the first term summarises all bioavailable forms of nitrogen the latter is dedicated to diazotrophs.

Line 52 what is a "dramatic" increase of N/P ratios?

Line 61 it should be avoided to merge the process of nitrification into budget considerations

Line 71ff If a microbial process like nitrification is a major scope of a study it should have been measured during the field work. Including these rates could improve the study significantly.

Line 103 are the detection limits indeed as reported? They seem high to me.

Line 133 the model has a depth resolution of 1.5m, the field data seem to have a spacing of 5-10m. This mismatch should be solved as the model validation can hardly be done with the data gathered.

Line 145 The authors may not use two seasons only to extrapolate to an entire year. Here the data of other studies could be used to generate a full annual data coverage.

Fig 2 and 4 have blanks. How are does ODV generate these? Are the gradients of riparian data too large?

Line 166 Here I do not agree. The nutrient concentrations in spring are highly variable from 15-5 micromol L-1.

Line 168 Fig 4 and 5 do not present any phosphate concentrations – the reference to the figures is not correct

Line 170ff average concentrations of all stations and depth have been calculated. Although I understand why this is done it makes of course no sense when a thermal stratification, a clear river plume and other features exist. Rephrasing and explaining this would help.

Line 230 Sv unit should use superscript

Line 244 there is a typo r=. . .

Line 251 trace amounts are usually much lower than 0.5 micromol per liter which is the detection limit given.

Line 270ff the assumption of similar nitrate fluxes in rivers without data based on the regional vicinity seems doubtful to me. Is the land use similar too?

Point 4.2.2 this paragraph tries to explain away all uncertainties and assumptions but the potential error is likely very high. As said above – this and the other fluxes need to

be treated using error estimates.

Point 4.2.3 was indeed the atmospheric deposition of entire China used? There should be tremendous differences across the country. May be I am misunderstanding something, but it seems that regional deposition data should be used. And again the uncertainty in the estimate needs to be included.

Line 326 unit

Page 18 and 19 the concerns explained above would need consideration to construct the budget differently

Line 363 is not a hypothesis but a well known fact

Line 456 delta as Greek letter. What about sediment resuspension and transport? Wouldn't that also blur any isotope signature?

Point 5 the conclusion would need a revision

---

## Author Comment (AC2) · 19 Apr 2021

**Referee's comments:**
Overall the topic is of interest and fits well to the scope of the journal Biogeosciences.
Moreover, a budget using the LOICZ approach was constructed using measurements but mainly data from other studies. The budget balances sources and sinks of nitrate and is supported by stable isotope data of nitrate. As a central conclusion the overwhelming role of nitrification as the major source of nitrate is presented.

**Authors' reply:**
    We thank the referee for his/her ideas on improving the manuscript and appreciate the careful review. In our opinion, the reviewer may not have appreciated the power of combined mass- and isotope budgets, which add a completely new dimension to standard mass-only budgets due to the process-specific isotope fractionation in the nitrogen cycle. This and an apparently poor choice of title on our side may have led to misunderstanding, which we hope to remedy in a revision. We address this point and others raised by the referee below.

**Referee's comments:**
The most significant error is the definition of equations (1) and (2). Sources and sinks are listed and supposed to be balanced. However, the source terms list nitrification and the loss terms list sedimentation. Both of these do not fulfill the criteria of a source or sink, respectively.
Nitrification is neither a source nor a sink for nitrate but simply a microbial process that converts ammonium via nitrite to nitrate. Nitrification does not generate new DIN for a system simply because the substrate of the nitrification process is ammonium and comes from internal turnover processes of organic matter. The LOICZ report (no 5 LOICZ BIOGEOCHEMICAL Modelling Guidelines, 1996) states "The important point to note with this reaction (nitrification) is that **carbon and phosphorus are not directly involved in the net reaction**. Again this makes the point that the relationship between NO3 and NH4 may be considered an "internal cycle" which need not be dealt with directly."

**Authors' reply:**
    We are sorry that our approach is apparently open to misunderstanding, and indeed will change the title of our manuscript to "**A nitrate budget of the Bohai Sea based on an isotope mass balance model**" in order to make our focus on reactive nitrogen clearer. Because anthropogenic impacts on biogeochemical cycles of marginal seas is always seen in amplified inputs of reactive N, we focus on this cycle. The budget in the manuscript thus is basically a budget of nitrate ion in the water mass, neither organic particles nor the sediment are included as active compartments. Instead, we expand the mass-based budget with an isotope-based budget to employ the added possibilities of dual nitrate isotopes to quantify nitrate sources and sinks. That is why

nitrification and sedimentation are considered as sources and sinks, because they affect the isotope budget.

The "LOICZ approach" that the reviewer refers to links water- and salt-balance to construct carbon, nitrogen and phosphorus budget models and is an established methodology to standardize mass flux estimates of these biogeochemically important elements in coastal systems on regional to local scales (D.C. Gordon et al., 1996;Smith et al., 2005). The underlying box model approach diagnoses water, salt and CNP-fluxes, for example to decide if systems are autotrophic (production exceeds respiration) or heterotrophic (respiration exceeds production) on the basis of deviations from stoichiometric Redfield ratios. We do not aim to decide if Bohai Sea nutrient cycles indicate net autotrophy or net heterotrophy, and thus specifically do not aim to include the carbon balance associated with the LOICZ approach. We establish a more generic box model approach specifically of the reactive nitrogen pools to provide mass flux estimates of inputs and outputs to the nitrate pool. Our approach thus goes beyond a mass flux estimate (as done in previous LOICZ-type budgets for Bohai Sea, e.g., Zhang et al. (2004)) by constraining some of the branches of reactive nitrogen in this coastal sea based on the tell-tale changes in dual isotope composition of nitrate, and the dual isotopic properties of sources and sinks. This approach thus differs fundamentally from the solely mass-balance approach that have been done previously and that have large uncertainties specifically in the internal nitrogen turnover. In our opinion the approach taken in our study is significantly more specific and is diagnostic of several important pathways of reactive nitrogen.

Nitrification is indeed an "internal" source, as the referee says: "nitrification does not generate new DIN for a system simply because the substrate of the nitrification process is ammonium and comes from internal turnover processes of organic matter", the initial source of newly nitrified nitrate is ammonium or organic matter which releases ammonium. However, nitrification affects the stable isotopic ratios and therefore, has to be included into our budget. For our study, N in organic matter is not involved in our box model because it is simply not present as nitrate ions.

**Referee's comments:**
Sedimentation is also a problematic variable for a balanced budget because it is not a removal process for nitrate. Organic material in sediments is prone to remineralization and resuspension. Only if that organic matter is permanently buried it can be considered a long-term sink. The finding that nitrification in the Bohai Bay is a very active process – generating large quantities of nitrate - is not a surprise but rather typical for coastal eutrophied systems.

**Authors' reply:**
We fully agree with the referee that "only if that organic matter is permanently buried it can be considered a long-term sink". However, sedimentation in our model is defined as the nitrate removal from the water pool:

because the process converts the nitrate ion to particulate nitrogen (PN), it is a sink for inorganic N and importantly, it also is associated with isotope fractionation. We only need to consider the net amount of newly produced PN, namely the amount of consumed nitrate in our model.

The significance of nitrification may be unsurprising to the reviewer, but in terms of environmental management and legislation, it makes a lot of difference whether a nitrate "target threshold concentration" can be controlled by reducing river or atmospheric inputs, or whether the nitrate is generated by sources that cannot be controlled (as is the case in Bohai Sea nitrification – the largest nitrate source).

**Referee's comments:**
Another problem I have with this manuscript is that it actually consists of two independent stories; one is the field study of nitrogen compounds and stable isotopes in the Bohai Sea and the other the budget. Both are rather disconnected although the authors try to include some measurements in the budget. The field data are rather distracting from the main scope because they suggest that part of the budget is based on measurements although, most data are taken from other publications. Of course the authors write very clearly where the numbers for the budget are from and which underlying assumptions were applied to derive mean values. Nevertheless, the field data and budget remain two different stories.

**Authors' reply:**
The main objective of this manuscript is the budget of mass fluxes constrained by the isotope budget, and indeed both field data and literature data are used for completing the budget. The basic data are obtained through our investigations, including the basic hydrology data (salinity and temperature), nutrients and nitrate isotopes, the properties of the particles and sediments. They are the bulk of the data used here and constitute the backbone of our study. They cannot be excluded if the whole budget needed to be constrained.

**Referee's comments:**
But data from only two seasons can hardly be used for a budget averaging annual mean fluxes.

**Authors' reply:**
Using data from two seasons is not ideal to extrapolate to the annual situation, but our data set brackets the intra-annual variability in the Bohai Sea. That marginal sea and the eastern Chinese seas in general are dominated by the monsoon circulation that imposes characteristic end-member states in summer and winter driven by opposite directions of monsoon. During our cruises, the early spring pattern was that of the winter season, as reflected by the vertically mixed water column, and results of the second seasonal sampling images a typical summer situation when sea water temperature was quite high and biological activity has consumed most of the nutrients. The annual situation thus is represented by the two most typical seasons. Likewise, Yellow River, the

most important external riverine source of nitrate, was monitored over 5 months in order to register the dry and flood seasons.

**Referee's comments:**
The results of the HAMSON model were not used at all and in the discussion the field data are only briefly mentioned.
**Authors' reply:**
  The HAMSOM model is described in the method part, and as stated in the text we use the modeled water transports only to calculate the "net export" of nitrate from Bohai Sea to the Yellow Sea in our mass- and isotope balance model. Because our main purpose was to estimate transports and not to delve into details of hydrodynamic circulation, we only gave a brief text and do not see the point of expanding the manuscript.

**Referee's comments:**
The final major concern is the lacking error estimate of the budget. All field data are subject to some degree of major or minor inaccuracy, which is not analyzed and not included in the budget calculations. Point 4.3.3 is insufficient and only addresses single sources. What's needed is an error propagation estimate.
**Authors' reply:**
  The errors coming from the measurement uncertainties could be done in more detail, and we will report the numbers in our revision. The errors of fluxes from the mass- and isotope balance are given as ranges in the manuscript.

**The specific reply for the small issues.**
L33: Reactive nitrogen is different to fixed nitrogen. While the first term summarises all bioavailable forms of nitrogen the latter is dedicated to diazotrophs.
**Authors' reply:**
Thank you and we will rephrase like "Reactive ($N_r$) is an essential nutrient of life on earth".

L52: what is a "dramatic" increase of N/P ratios?
**Authors' reply:** N/P ratio increased about 30 times, we now give this number.

L61: it should be avoided to merge the process of nitrification into budget considerations
**Authors' reply:** As explained above, we were trying to constrain a model of nitrate in the water of Bohai Sea, and even though nitrification is a internal cycling process, it leaves a significant imprint on the isotope balance.

L71: If a microbial process like nitrification is a major scope of a study it should have been measured during the field work. Including these rates could improve

the study significantly.

**Authors' reply:** The nitrification rates in both water column and sediments of Chinese marginal seas are not well documented, and our manuscript remedies that lack of information. We fully agree with referee's suggestion that one way to estimate nitrification the direct measurements of nitrification by incubation experiments (Ward, 2011). A second widely used approach (Wankel et al., 2007;DiFiore et al., 2009;Sigman et al., 2009) is mass- and isotope-based modeling that we use here.

L103: are the detection limits indeed as reported? They seem high to me.

**Authors' reply:** Thank you for catching this error: the detection limit for $NO_x$ is 0.05 µmol $L^{-1}$.

L133: the model has a depth resolution of 1.5m, the field data seem to have a spacing of 5-10m. This mismatch should be solved as the model validation can hardly be done with the data gathered.

**Authors' reply:** The upper 50 m of the HAMSOM model are resolved by layers of 5 m thickness. So, this coincides nicely with the resolution of our observational data. HAMSOM has been applied in the Bohai Sea since last century, we believe the HAMSOM was validated good in the Bohai Sea (Jia and Chen, 2021;Hainbucher et al., 2004;Huang et al., 1999).

L145: The authors may not use two seasons only to extrapolate to an entire year. Here the data of other studies could be used to generate a full annual data coverage.

**Authors' reply:** As mentioned above, the data from two seasons is not ideal to present the annual situation, but as we discussed in the manuscript, the Bohai Sea or even the eastern Chinese seas are monsoon-driven systems where most different seasons are summer and winter with opposite directions of monsoon. The isotope data are crucial for our approach and literature data of nitrate isotopes have to our knowledge not been published in Bohai Sea so far.

Fig 2 and 4 have blanks. How are does ODV generate these? Are the gradients of riparian data too large?

**Authors' reply:** ODV users can display their data in gridded form with the calculation method called "weighted-average gridding". Users can choose the extension scale of each data point, so that the blanks are places without interpolated values. By increasing the horizontal or vertical range of each datum the blanks can be filled if necessary, but the blanks do in our opinion not obscure the patterns.

Line 166 Here I do not agree. The nutrient concentrations in spring are highly variable from 15-5 micromol L-1.

**Authors' reply:** Here we will rephrase to: "Nutrient concentrations in spring

were almost vertically uniform, consistent with temperatures and salinities, and no distinct nutricline was observed".

Line 168 Fig 4 and 5 do not present any phosphate concentrations – the reference to the figures is not correct
**Authors' reply:** Nitrate or dissolved inorganic nitrogen is our key point, so that we decided against showing phosphate profiles. We will delete the reference to figures.

Line 170ff average concentrations of all stations and depth have been calculated. Although I understand why this is done it makes of course no sense when a thermal stratification, a clear river plume and other features exist. Rephrasing and explaining this would help.
**Authors' reply:** We fully realize that heterogeneous distribution of any parameters introduces errors. Thus, the patterns were briefly described and displayed as graphs to inform the reader.

Line 230 Sv unit should use superscript
**Authors' reply:** We will correct it.

Line 244 there is a typo r=. . .
**Authors' reply:** We will correct it.

Line 251 trace amounts are usually much lower than 0.5 micromol per liter which is the detection limit given.
**Authors' reply:** Thanks again for noticing this typo, it should be 0.05 µmol $L^{-1}$ which is closer to the "trace amount".

Line 270ff the assumption of similar nitrate fluxes in rivers without data based on the regional vicinity seems doubtful to me. Is the land use similar too?
**Authors' reply:** In the manuscript, the nitrate concentrations of Shuangtaizi River, Daling River, and Xiaoling River were set to be the same as the Daliao River. Shuantaizi River and Daliao River are quite near each other, their drainage basins are similarly populated and industrialized. Daling River and Xiaoling River are also close and both polluted by human activities(Wang et al., 2010). The nitrate concentration of these four rivers was reported to be similar (Zhang et al., 2007). Although all are quite polluted rivers, their water discharges are relatively small and only account for 4.1% of the discharge by the 8 biggest rivers into the Bohai Sea. Our estimate of nitrate concentrations for the three rivers is thus justified and erroneous estimates would only introduce small errors.

Point 4.2.2 this paragraph tries to explain away all uncertainties and assumptions but the potential error is likely very high. As said above – this and

the other fluxes need to be treated using error estimates.

**Authors' reply:** The error estimates could be done.

Point 4.2.3 was indeed the atmospheric deposition of entire China used? There should be tremendous differences across the country. May be I am misunderstanding something, but it seems that regional deposition data should be used. And again the uncertainty in the estimate needs to be included.

**Authors' reply:** We realise that the estimates may be problematic as well, but the results calculated by using different field data agree well and we are not aware of better data. The measurement-based estimates in the Bohai Sea are $3.4 \times 10^9$ mol yr$^{-1}$ (Zhang et al., 2004) and $3.1 \times 10^9$ mol yr$^{-1}$ (Liu et al., 2003), respectively. Other indirect estimates amount to $3.7 \times 10^9$ mol yr$^{-1}$ (Kim et al., 2019) and $3.6 \times 10^9$ mol yr$^{-1}$ (Zhao et al., 2017), respectively. We decided to use the number $3.4 \times 10^9$ mol yr$^{-1}$ because it was measured directly and is in the middle range of these estimates.

Line 326 unit

**Authors' reply:** We will correct it.

Page 18 and 19 the concerns explained above would need consideration to construct the budget differently

**Authors' reply:** We will make clear that we calculate the budget of "nitrate" instead of all reactive nitrogen.

Line 363 is not a hypothesis but a well known fact

**Authors' reply:** We will change the text "The sources of nitrate for BHS are river inputs, submarine fresh groundwater input, atmospheric deposition, and remineralization."

Line 456 delta as Greek letter. What about sediment resuspension and transport? Wouldn't that also blur any isotope signature?

**Authors' reply:**

Our data show that the $\delta^{15}N$ values of sediment increase from the Yellow River estuary to the Bohai Strait along the pathway of water and particle transport (4.55‰-5.58‰) and mirror a decrease in terrestrial particles with increasing distance from the estuary. Importantly, the terrestrial signal of Yellow River particles disappears in short distance from the estuary. Furthermore, the isotope data suggest that resuspension of sediment with terrestrial signature is not significant.

Point 5 the conclusion would need a revision

**Authors' reply:** Thanks and will revise.

**References:**

D.C. Gordon, J., Boudreau, P. R., Mann, K. H., Ong, J.-E., Silvert, W. L., Smith, S. V., Wattayakorn, G., Wulff, F., and Yanagi, T.: LOICZ BIOGEOCHEMICAL MODELLING GUIDELINES, LOICZ REPORTS & STUDIES, 5, 1996.

DiFiore, P. J., Sigman, D. M., and Dunbar, R. B.: Upper ocean nitrogen fluxes in the Polar Antarctic Zone: Constraints from the nitrogen and oxygen isotopes of nitrate, Geochem. Geophys. Geosyst., 10, https://doi.org/10.1029/2009GC002468, 2009.

Hainbucher, D., Hao, W., Pohlmann, T., Sündermann, J., and Feng, S.: Variability of the Bohai Sea circulation based on model calculations, J. Mar. Syst., 44, 153-174, https://doi.org/10.1016/j.jmarsys.2003.09.008, 2004.

Huang, D., Su, J., and Backhaus, J. O.: Modelling the seasonal thermal stratification and baroclinic circulation in the Bohai Sea, Cont. Shelf Res., 19, 1485-1505, https://doi.org/10.1016/S0278-4343(99)00026-6, 1999.

Jia, B., and Chen, X. e.: Application of an ice-ocean coupled model to Bohai Sea ice simulation, Journal of Oceanology and Limnology, 39, 1-13, 10.1007/s00343-020-9168-8, 2021.

Kim, H., Park, G.-H., Lee, S.-E., Kim, Y.-i., Lee, K., Kim, Y.-H., and Kim, T.-W.: Stable isotope ratio of atmospheric and seawater nitrate in the East Sea in the northwestern Pacific ocean, Mar. Pollut. Bull., 149, 110610, https://doi.org/10.1016/j.marpolbul.2019.110610, 2019.

Liu, S. M., Zhang, J., and Jiang, W. S.: Pore water nutrient regeneration in shallow coastal Bohai Sea, China, J. Oceanogr., 59, 377-385, https://doi.org/10.1023/A:1025576212927, 2003.

Sigman, D. M., DiFiore, P. J., Hain, M. P., Deutsch, C., Wang, Y., Karl, D. M., Knapp, A. N., Lehmann, M. F., and Pantoja, S.: The dual isotopes of deep nitrate as a constraint on the cycle and budget of oceanic fixed nitrogen, Deep-Sea Res. Pt. I, 56, 1419-1439, https://doi.org/10.1016/j.dsr.2009.04.007, 2009.

Smith, S. V., Buddemeier, R. W., Wulff, F., Swaney, D. P., Camacho-Ibar, V. F., David, L. T., Dupra, V. C., Kleypas, J. A., San Diego-McGlone, M. L., McLaughlin, C., and Sandhei, P.: C, N, P Fluxes in the Coastal Zone, in: Coastal Fluxes in the Anthropocene: The Land-Ocean Interactions in the Coastal Zone Project of the International Geosphere-Biosphere Programme, edited by: Crossland, C. J., Kremer, H. H., Lindeboom, H. J., Marshall Crossland, J. I., and Le Tissier, M. D. A., Springer Berlin Heidelberg, Berlin, Heidelberg, 95-143, 2005.

Wang, H., Li, Z., Lei, K., and Zhang, Z.: Analysis of Driving Forces and Variations of the Runoff and Suspended Sediment Discharge of the Daling and Xiaoling Rivers into the Sea over the Past Twenty Years, Research of Environmental Sciences (in Chinese), 23, 1236-1242, 2010.

Wankel, S. D., Kendall, C., Pennington, J. T., Chavez, F. P., and Paytan, A.: Nitrification in the euphotic zone as evidenced by nitrate dual isotopic composition: Observations from Monterey Bay, California, Global Biogeochem. Cycles, 21, https://doi.org/10.1029/2006GB002723, 2007.

Ward, B. B.: Chapter thirteen - Measurement and Distribution of Nitrification Rates in the Oceans, in: Methods Enzymol., edited by: Klotz, M. G., Academic Press, 307-323, 2011.

Zhang, J., Yu, Z., Raabe, T., Liu, S., Starke, A., Zou, L., Gao, H., and Brockmann, U.: Dynamics of inorganic nutrient species in the Bohai seawaters, J. Mar. Syst., 44, 189-212, https://doi.org/10.1016/j.jmarsys.2003.09.010, 2004.

Zhang, J., XIa, B., Gui, Z., and Jiang, C.: Contaminative Conditions Evaluation of Sixteen Main Rivers Flowing into Sea Around Bohai Sea, in Summer of 2005, Environmental Science (in Chinese), 28, 2409-2415, 2007.

Zhao, Y., Zhang, L., Chen, Y., Liu, X., Xu, W., Pan, Y., and Duan, L.: Atmospheric nitrogen deposition to China: A model analysis on nitrogen budget and critical load exceedance, Atmos. Environ., 153, 32–40, https://doi.org/10.1016/j.atmosenv.2017.01.018, 2017.

---

## Short Comment (SC1) · 7 Aug 2021

Review comments This manuscript aims to clarify the N sources and N cycling on the Bohai Sea, by investigating the NO3 concentration and its dual isotope variations and other chemical parameters to distinguish reactive of fixed nitrogen sources and provide an insight into the N biogeochemical transformations in the Bohai Sea, using a combined mass and isotope balance model. This topic is important in Bohai Sea (BHS) region for understanding the N biogeochemical process for marine environment management and recovering. It is an interesting manuscript with heated topics. I think the discussion is sufficient and the results is quite interesting. The Manuscript contains several typos and grammatical errors and the author is suggested to carefully check the

manuscript such as reference, hard to read sentences. This manuscript needs some improvements. My decision is accepting with minor revision regarding the following review comments. More specific and general comments are given below.

Review 1. In line 13, 'The Bohai Sea' seems to be appeared first time, It is suggested to use 'The Bohai Sea(BHS)' instead here.

2. The author is suggested to check several sentences which is difficult to read such as In line 15, 'It is therefore crucial to quantify the reactive nitrogen input to the BHS and to understand the processes and determine the quantities of nitrogen eliminated in and exported from the BHS' is suggested to revise as 'Therefore, it is crucial to quantify the reactive nitrogen input to the BHS and understand the processes and determine the quantities of nitrogen eliminated in and exported from the BHS.'

3. The author is suggested to check the whole manuscript about some small mistakes, such as in line 44-45 Chen, 2009 should be Chen et al., 2009, Su, 2001 could not find in the reference list.

4. In line 24-25, In here, Ground water should be groundwater, it is suggested that "submarine discharge of nitrate with fresh ground water' changed to 'submarine fresh groundwater discharge of nitrate'.

5. In line 23-26, 'The main nitrogen sources are rivers contributing 17.5%-20.6% and the combined terrestrial runoff (including submarine discharge of nitrate with fresh ground water) accounting for 22.6%-26.5% of the nitrate input to the BHS while atmospheric input contributes only 6.3%-7.4% to total nitrate.' In here, firstly you discussed about nitrogen sources, then mentioned about nitrate percentage. It seems a little confuse, please use nitrate or nitrogen (DTN? ) instead.

6. In line 29-30, the sentence' A further eutrophication of the BHS could, however, induce water column hypoxia and denitrification as already observed – often seasonally off river mouths - in other marginal seas.' is hard to read, please revise it as more simple

way.

7. In line 43 (Smith et al., 2003;Liu et al., 2009). Please make sure if a space is needed between two citations.

8. In line 70, a comma is needed after study. For this study, we analyzed. . . . . .

9. In line 73, Aim of the study - The aim of the study.

10. In line 75, . . .et al., 2011), Please remove the comma here.

11. In line 92, The author described as 'Samples were taken monthly in May, July to November from Yellow River, and in November from Daliao River, Hai River, Luan River and Xiaoqing River (Fig. 1).', Why the water sample is only taken monthly from summer to winter in Yellow river(Also why it is not taken in June?), the other river is only a winter sample, Because it contains dry and wet season in the research region, is it enough to calibrate/validate the mass balance model using one month data?

12. In line 135-140, the author described the model by using HAMSOM, and calculate the model in year 2018. How about the warm up periods of the model? And how about the calibration/validation process, the author is suggest to describe the model more detaily.

13. In line 165-176, Please uniform the nutrient name in this part, Such as there are NH4+ in the text but NH4+ -N in the figure.

14. In line 247, 'nitrate- rich' please remove the space before rich.

15. In line 250, '(= halo- and nutricline)' I did not understand the expression here, Could the author explain it more clearly?

16. In line 245, the author described as 'the YR is one of the major sources of these nutrients in the BHS' but not discussed the nutrient contents from other rivers, The author is suggested to described more detaily here.

17. In line 249, '(see the discussion of chapter 4.2.5).' I am not sure if it is ok to refer as this way. Because it makes the reader more confused about the discussion part.

18. In line 314, 'sea water.' In manuscript, there are two descriptions as 'sea water' and 'seawater', please uniform the callings.

19. In line 317, 'north China Plain' should be 'North China Plain'.

20. In line 327, 'The difference of the' of – between.

21. In line 340, 'from sediment' to ''from the sediment.

22. In line 345, 'Sinking particles in the BHS have a $\delta$15N of 5.2‰ ($\delta$15Nsink),' I am confused about this part, Is this data measured from this curies? This suspended particulate matter value is not shown in the manuscript(in Line 200-206, it shows The average $\delta$15N of SPM in spring was 4.8$\pm$0.9‰, The author is suggested to add reference or method of this data.

23. In line 363-365, 'ground water' should be 'groundwater' 'Most important sinks' should be 'The most important sinks' 'steady state' should be 'steady-state'.

24. In line 371, 'deposited nitrate„' please remove a comma.

25. In line 440, 'in the eastern Hainan Island' remove 'the'.

---

## Referee Comment (RC2) · Anonymous Referee #2 · 8 Aug 2021

Tian et al. used the isotope mass balance model to characterize and quantify reactive nitrogen sources and sinks in the Bohai Sea through the measurements of nutrient, nitrate $\delta$15N and $\delta$18O, and $\delta$15N of suspended matters and sediments. The authors used the results both from their work and previous studies trying to give a more comprehensive estimation of nitrate sources and sinks from various end members. This work would improve the understanding of N cycle in the Bohai Sea, a typical semi-enlosed bay influenced by anthropogenic nitrogen input, I think this manuscript could be accepted after a minor revision. Here, I have some specific comments for this study: 1. I think this study would need a little bit more detail discussing of the model uncertainties. There could be some uncertainties in this isotope mass balance mode due to many assumptions in this study. For example, there are many assumptions for

using the end member of sedimentation (section 4.2.5). As the isotope fractionation associated with the processes of assimilation and nitrification is complicated, I think it may not be suitable to give fixed values of $\delta$15N and $\delta$18O to the correlated end members. I suggest to give varying values of $\delta$15N and $\delta$18O with reasonable range when applying to the isotope mass balance mode. 2. In addition, in summer, nitrate was almost depleted in the most area of the Bohai Sea, suggesting an enhanced photosynthesis rate and assimilation rate in this season. The residual nitrate would have high $\delta$15N and $\delta$18O values. It may need to evaluate rationality by adopting average values of nitrate concentrations, $\delta$15N and $\delta$18O in the two seasons when applying to the isotope mass balance model.

Please also note the supplement to this comment:
https://bg.copernicus.org/preprints/bg-2020-471/bg-2020-471-RC2-supplement.pdf

---

## Author Comment (AC3) · 24 Aug 2021

Review

*Tian et al. used the isotope mass balance model to characterize and quantify reactive nitrogen sources and sinks in the Bohai Sea through the measurements of nutrient, nitrate δ15N and δ18O, and δ15N of suspended matters and sediments. The authors used the results both from their work and previous studies trying to give a more comprehensive estimation of nitrate sources and sinks from various end members. This work would improve the understanding of N cycle in the Bohai Sea, a typical semi-enclosed bay influenced by anthropogenic nitrogen input, I think this manuscript could be accepted after a minor revision. Here, I have some specific comments for this study:*

We thank the reviewer for a positive and encouraging review and thoughtful comments and queries. In the following we address the points raised and how they will be implemented in the revision.

*1. I think this study would need a little bit more detail discussing of the model uncertainties. There could be some uncertainties in this isotope mass balance mode due to many assumptions in this study. For example, there are many assumptions for using the end member of sedimentation (section 4.2.5). As the isotope fractionation associated with the processes of assimilation and nitrification is complicated, I think it may not be suitable to give fixed values of $\delta^{15}N$ and $\delta^{18}O$ to the correlated end members. I suggest to give varying values of $\delta^{15}N$ and $\delta^{18}O$ with reasonable range when applying to the isotope mass balance mode.*

**Authors' reply:** The reviewer is of course correct in pointing out the errors arising from adopting fixed end member values. We will set up the uncertainties for the end members in the revised manuscript.

*2. In addition, in summer, nitrate was almost depleted in the most area of the Bohai Sea, suggesting an enhanced photosynthesis rate and assimilation rate in this season. The residual nitrate would have high $\delta^{15}N$ and $\delta^{18}O$ values. It may need to evaluate rationality by adopting average values of nitrate concentrations, $\delta^{15}N$ and $\delta^{18}O$ in the two seasons when applying to the isotope mass balance model.*

**Authors' reply:** Thanks for this suggestion that we will implement in the revision (see supplement below). As mentioned in the manuscript, only a subset of samples could be analyzed due to the low nitrate concentrations in summer, and most of these are from the Yellow River Diluted Water that had $[NO_3^-]$ >1.7µmol/L. The average values of $\delta^{15}N$ and $\delta^{18}O$ of the Bohai Sea in summer were 9.9±3.5‰ (n=23) and 8.7±3.3‰ (n=23). Although measurements could not constrain the range of nitrate isotope values, the lacking isotope data can be roughly estimated:

According to the T-S patten in summer, the Bohai Sea water can be considered as a two-end member mixture between fresh water discharged from Yellow River (YR) and sea water of central Bohai Sea, the nitrate concentration only affected by physically mixing hence can be calculated (Supplement 1). The isotope effect of assimilation for nitrate in the Bohai Sea follows the "steady-state model" rather than the Rayleigh model because the Yellow River supplies nitrate continuously (Sigman and Fripiat, 2019). Thus, the estimated dual nitrate isotope values can be calculated according to equation (1) and (2):

$$\delta^{15}N_{reactant} = \delta^{15}N_{initial} + {}^{15}\varepsilon(1-f) \ (1)$$
$$\delta^{18}O_{reactant} = \delta^{18}O_{initial} + {}^{18}\varepsilon(1-f) \ (2)$$

In Eq.1 and Eq. 2, $f$ is equal to the observed nitrate concentration divided by of result of the twoend member model, $\delta^{15}N_{initrial}$ is equal to the end member of YR, and $\delta^{15}N_{reactant}$ is the estimated value of the residual nitrate, the value we need. The average of $^{15}\varepsilon$ and $^{18}\varepsilon$ adopted here are 5‰ (Granger et al., 2010; DiFiore et al., 2009; Liu et al., 2017; Wu et al., 2019; Umezawa et al., 2013; Wang et al., 2016).

The readjusted values of $\delta^{15}N$ and $\delta^{18}O$ for the Bohai Sea in summer is 12.8±2.7‰ (n=85) and 9.1±1.9‰ (n=85), respectively, resulting in seasonally averaged values of $\delta^{15}N$ and $\delta^{18}O$ of 10.3‰ and 10.6‰, respectively. These values induce about 5%-9% deviations of the mass fluxes in our box model. Because this estimate is also based on the two-end member mixing model and isotopic fractionation equations, we think that this part probably is better placed in the uncertainty discussion that will be included in the revision.

**Supplement 1 The estimate of two end member mixing of nitrate**

The YR provides warm, fresh and nitrate enriched water whereas cold, saline and nitrate depleted water was observed near the area of the outer Liaodong Bay in both spring and summer. Thus, there were two end members to be considered in a mixing model. One should be aware that a contribution of atmospheric nitrogen is included in the marine end member as well.

[Figure]

Fig. S1 Temperature vs. salinity in Bohai Sea in spring (left) and summer (right). The values adopted for the two nitrate end members were mainly based on this pattern

The values of these two end members are shown in Table S-1. The summer basic pattern of temperature and salinity was similar to that of spring. Thus, the fraction of water originating from YR and the BHS during the mixing process can be calculated follow (1) and (2):

$$S = S_r \times f_r + S_s \times f_s \quad (1)$$
$$f_r + f_s = 1 \quad (2)$$

where S, $S_r$ and $S_s$ refers to the observed salinity in study area, the end member value of river and sea, respectively. $f_r$ and $f_s$ refers to the fraction of river and sea water, respectively. The modeled nitrate concentration and modeled $\delta^{15}N$ and $\delta^{18}O$ values can be calculated following equations (3), (4) and (5):

$$[NO_3^-]_m = [NO_3^-]_r \times f_r + [NO_3^-]_s \times f_s \quad (3)$$
$$\delta^{15}N_m[NO_3^-]_m = \delta^{15}N_r[NO_3^-]_r + \delta^{15}N_s[NO_3^-]_s \quad (4)$$
$$\delta^{18}O_m[NO_3^-]_m = \delta^{18}O_r[NO_3^-]_r + \delta^{18}O_s[NO_3^-]_s \quad (5)$$

where $[NO_3^-]_m$, $[NO_3^-]_r$ and $[NO_3^-]_r$ refers to the modeled nitrate concentration and the end member nitrate concentration values of river and sea, respectively. $\delta^{15}N_m/\delta^{18}O_m$, $\delta^{15}N_r/\delta^{18}O_r$ and $\delta^{15}N_s/\delta^{18}O_s$ refer to the modeled $\delta^{15}N$ and $\delta^{18}O$ values, and the end member $\delta^{15}N$ and $\delta^{18}O$ values of river and sea, respectively.

Table S-1 Two end member values in Bohai Sea

|  |  | Riverine | Marine |
|---|---|---|---|
| spring | Salinity | 29.9 | 33.0 |
|  | Nitrate/μmol/L | 31.1 | 6.0 |
|  | $\delta^{15}N‰$ | 9.5 | 6.0 |
|  | $\delta^{18}O‰$ | 6.8 | 12.5 |
| summer | Salinity | 28.5 | 32.5 |
|  | Nitrate/μmol/L | 13.6 | 2.0 |
|  | $\delta^{15}N‰$ | 9.9 | 9.5 |
|  | $\delta^{18}O‰$ | 5.3 | 8.2 |

**References:**

DiFiore, P. J., Sigman, D. M., and Dunbar, R. B.: Upper ocean nitrogen fluxes in the Polar Antarctic Zone: Constraints from the nitrogen and oxygen isotopes of nitrate, Geochem. Geophys. Geosyst., 10, https://doi.org/10.1029/2009GC002468, 2009.

Granger, J., Sigman, D. M., Rohde, M., Maldonado, M., and Tortell, P.: N and O isotope effects during nitrate assimilation by unicellular prokaryotic and eukaryotic plankton cultures, Geochim. Cosmochim. Acta, 74, 1030-1040, https://doi.org/10.1016/j.gca.2009.10.044, 2010.

Liu, S. M., Altabet, M. A., Zhao, L., Larkum, J., Song, G. D., Zhang, G. L., Jin, H., and Han, L. J.: Tracing Nitrogen Biogeochemistry During the Beginning of a Spring Phytoplankton Bloom in the Yellow Sea Using Coupled Nitrate Nitrogen and Oxygen Isotope Ratios, J. Geophys. Res.-Biogeo., 122, 2490-2508, 10.1002/2016jg003752, 2017.

Sigman, D. M., and Fripiat, F.: Nitrogen Isotopes in the Ocean, in: Encyclopedia of Ocean Sciences (Third Edition), edited by: Cochran, J. K., Bokuniewicz, H. J., and Yager, P. L., Academic Press, Oxford, 263-278, 2019.

Umezawa, Y., Yamaguchi, A., Ishizaka, J., Hasegawa, T., Yoshimizu, C., Tayasu, I., Yoshimura, H., Morii, Y., Aoshima, T., and Yamawaki, N.: Seasonal shifts in the contributions of the Changjiang River and the Kuroshio Current to nitrate dynamics at the continental shelf of the northern East China Sea based on a nitrate dual isotopic composition approach, Biogeosci. Disc., 10, 10143-10188, https://doi.org/10.5194/bg-11-1297-2014, 2013.

Wang, W., Yu, Z., Song, X., Wu, Z., Yuan, Y., Zhou, P., and Cao, X.: The effect of Kuroshio Current on nitrate dynamics in the southern East China Sea revealed by nitrate isotopic composition, J. Geophys. Res.-Oeans, 121, 7073-7087, https://doi.org/10.1002/2016JC011882, 2016.

Wu, Z., Yu, Z., Song, X., Wang, W., Zhou, P., Cao, X., and Yuan, Y.: Key nitrogen biogeochemical processes in the South Yellow Sea revealed by dual stable isotopes of nitrate, Estuar. Coast. Shelf Sci., 225, 106222, https://doi.org/10.1016/j.ecss.2019.05.004, 2019.

---

## Author Comment (AC4) · 24 Aug 2021

Review SC1

We thank the reviewer for helpful comments and corrections. In the following we address each of the points raised:

*1. In line 13, 'The Bohai Sea' seems to be appeared first time, It is suggested to use 'The Bohai Sea(BHS)' instead here.*
**Authors' reply:** We will correct this.

*2. The author is suggested to check several sentences which is difficult to read such as In line 15, 'It is therefore crucial to quantify the reactive nitrogen input to the BHS and to understand the processes and determine the quantities of nitrogen eliminated in and exported from the BHS' is suggested to revise as 'Therefore, it is crucial to quantify the reactive nitrogen input to the BHS and understand the processes and determine the quantities of nitrogen eliminated in and exported from the BHS.'*
**Authors' reply:** We will correct it.

*3. The author is suggested to check the whole manuscript about some small mistakes, such as in line 44-45 Chen, 2009 should be Chen et al., 2009, Su, 2001 could not find in the reference list.*
**Authors' reply:** We checked the "Chen, 2009" reference again and the sole author of this paper is Chen-Tung Arthur Chen (please see the link here:
https://www.sciencedirect.com/science/article/abs/pii/S0924796309000748 ).
The reference "Su, 2001" is called up in line 642 and the citation was indeed omitted - will be added. Also, the paper will be thoroughly checked for small mistakes.

*4. In line 24-25, In here, Ground water should be groundwater, it is suggested that "submarine discharge of nitrate with fresh ground water' changed to 'submarine fresh groundwater discharge of nitrate'.*
**Authors' reply:** We will correct this as proposed.

*5. In line 23-26, 'The main nitrogen sources are rivers contributing 17.5%-20.6% and the combined terrestrial runoff (including submarine discharge of nitrate with fresh ground water) accounting for 22.6%-26.5% of the nitrate input to the BHS while atmospheric input contributes only 6.3%-7.4% to total nitrate.' In here, firstly you discussed about nitrogen sources, then mentioned about nitrate percentage. It seems a little confuse, please use nitrate or nitrogen (DTN? ) instead.*
**Authors' reply:** In the revision, we will correct "The main nitrogen sources" in line 23 to "The main nitrate sources" to be more explicit.

*6. In line 29-30, the sentence' A further eutrophication of the BHS could, however, induce water column hypoxia and denitrification as already observed – often seasonally off river mouths - in other marginal seas.' is hard to read, please revise it as more simple way.*
**Authors' reply:** The restructured sentence could be: "However, a further eutrophication of the BHS could induce water column hypoxia and denitrification, as is increasingly observed in other marginal seas and seasonally off river mouths**".**

*7. In line 43 (Smith et al., 2003;Liu et al., 2009). Please make sure if a space is needed between two citations.*

**Authors' reply:** Spaces will be inserted between multiple citations in the revised version of the manuscript.

*8. In line 70, a comma is needed after study. For this study, we analyzed. . .. . .*

**Authors' reply:** We will correct this.

*9. In line 73, Aim of the study - The aim of the study.*

**Authors' reply:** We will correct it.

*10. In line 75, . . .et al., 2011), Please remove the comma here.*

**Authors' reply:** We will correct it.

*11. In line 92, The author described as 'Samples were taken monthly in May, July to November from Yellow River, and in November from Daliao River, Hai River, Luan River and Xiaoqing River (Fig. 1).', Why the water sample is only taken monthly from summer to winter in Yellow river(Also why it is not taken in June?), the other river is only a winter sample, Because it contains dry and wet season in the research region, is it enough to calibrate/validate the mass balance model using one month data?*

**Authors' reply:** The sampling for Yellow River (YR) was fit into the time schedule of the lead author, who participated in a ship sampling expedition to the Yellow Sea in June.

In our view, the data still are representative for the following reasons: The flood season of the YR is July to October (MWR, 2019), so that we have 4 months representative of the flood season (July, August, September and October) and 2 months for dry season (May and November). This means that our data set covers flood and dry seasons, although admittedly not in an ideal way. YR emptied $333.8 \times 10^9$ m$^3$ water and $8.0 \times 10^9$ mol nitrate to the Bohai Sea in 2018 and accounted for 85% and 84% of water and nitrate discharge among the largest rivers discharging to the Bohai Sea, respectively. The nitrate $\delta^{15}$N and $\delta^{18}$O of YR changed little during the sampling period. Because the average values of nitrate $\delta^{15}$N and $\delta^{18}$O for rivers are mass weighted instead of arithmetic mean values, the change of nitrate $\delta^{15}$N and $\delta^{18}$O for rivers with low nitrate discharges would induce little change of the average values.

For instance, the nitrate $\delta^{15}$N and $\delta^{18}$O of Daliao River in the flood season was reported as 20.1‰ and 9.4‰, respectively (Yue et al., 2013). Assuming that the flood season in Daliao River basin is also 4 months like in the YR basin, the flood and dry season mass weighted averaged nitrate $\delta^{15}$N and $\delta^{18}$O is 13.4‰ and 3.5‰, respectively. Combing these values with the rest of the rivers, the Daliao discharge resulted in quite small relative deviations (0.7% for $\delta^{15}$N and 0.8% for $\delta^{18}$O) to our data used in the manuscript ($\delta^{15}N_r$=10.0‰ and $\delta^{18}O_r$=1.3‰). We fully agree with the referee that the more completed monthly sampling for rivers will improve the results, but also consider the data at hand to be quite reliable.

*12. In line 135-140, the author described the model by using HAMSOM, and calculate the model in year 2018. How about the warm up periods of the model? And how about the calibration/validation process, the author is suggest to describe the model more detailly.*

**Authors' reply:** The spin-up period of this model is 1 year. The HAMSOM model has been applied to investigate the Bohai Sea physical circulation for several decades now and has been extensively validated in the Bohai Sea (Jia and Chen, 2021; Hainbucher et al., 2004; Huang et al., 1999). This information will be added to the manuscript.

*13. In line 165-176, Please uniform the nutrient name in this part, Such as there are NH4+ in the text but NH4+ -N in the figure.*
**Authors' reply:** The text in the figures will be corrected.

*14. In line 247, 'nitrate- rich' please remove the space before rich.*
**Authors' reply:** We will correct it.

*15. In line 250, '(= halo- and nutricline)' I did not understand the expression here, Could the author explain it more clearly?*
**Authors' reply:** The original text in the manuscript is "the water is stratified with the thermocline (= halo- and nutricline) at about 8 m water depth", we intended to tell the readers that the thermocline, halocline and nutricline were all observed at 8 m water depth. This will be rephrased.

*16. In line 245, the author described as 'the YR is one of the major sources of these nutrients in the BHS' but not discussed the nutrient contents from other rivers, The author is suggested to described more detailly here.*
**Authors' reply:** As we described above, Yellow River discharged $333.8 \times 10^9$ m$^3$ water and $8.0 \times 10^9$ mol nitrate to the Bohai Sea in 2018, accounting for 85% and 84% of water and nitrate discharge of all large rivers in the Bohai Sea, respectively. We will add this information to the revised version.

*17. In line 249, '(see the discussion of chapter 4.2.5).' I am not sure if it is ok to refer as this way. Because it makes the reader more confused about the discussion part.*
**Authors' reply:** The parentheses and the text included will be deleted.

*18. In line 314, 'sea water.' In manuscript, there are two descriptions as 'sea water' and 'seawater', please uniform the callings.*
**Authors' reply:** We will use "seawater" in the revised version of the manuscript.

*19. In line 317, 'north China Plain' should be 'North China Plain'.*
**Authors' reply:** We will correct it.

*20. In line 327, 'The difference of the' of – between.*
**Authors' reply:** We will correct it.

*21. In line 340, 'from sediment' to ''from the sediment.*
**Authors' reply:** We will correct it.

*22. In line 345, 'Sinking particles in the BHS have a $\delta^{15}$N of 5.2‰ ($\delta^{15}N_{sink}$),' I am confused about this part, is this data measured from this curies? This suspended particulate matter value is not*

*shown in the manuscript (in Line 200-206, it shows the average δ15N of SPM in spring was 4.8±0.9‰), The author is suggested to add reference or method of this data.*

**Authors' reply:** The $\delta^{15}N$ of particles in spring and summer was 4.8±0.9‰ (line 205) and 5.6±0.8‰ (line 207), respectively. The annually averaged value $\delta^{15}N_{sink}$ were calculated as the mean value of spring and summer. This will be described in the revised version.

*23. In line 363-365, 'ground water' should be 'groundwater' 'Most important sinks' should be 'The most important sinks' 'steady state' should be 'steady-state'.*

**Authors' reply:** We will correct it.

*24. In line 371, 'deposited nitrate,,' please remove a comma.*

**Authors' reply:** We will correct it.

*25. In line 440, 'in the eastern Hainan Island' remove 'the'.*

**Authors' reply:** We will correct it.

**References:**

Hainbucher, D., Hao, W., Pohlmann, T., Sündermann, J., and Feng, S.: Variability of the Bohai Sea circulation based on model calculations, J. Mar. Syst., 44, 153-174, https://doi.org/10.1016/j.jmarsys.2003.09.008, 2004.

Huang, D., Su, J., and Backhaus, J. O.: Modelling the seasonal thermal stratification and baroclinic circulation in the Bohai Sea, Cont. Shelf Res., 19, 1485-1505, https://doi.org/10.1016/S0278-4343(99)00026-6, 1999.

Jia, B., and Chen, X. e.: Application of an ice-ocean coupled model to Bohai Sea ice simulation, Journal of Oceanology and Limnology, 39, 1-13, 10.1007/s00343-020-9168-8, 2021.

MWR, C.: China River Sediment Bulletin 2018 (in Chinese), Beijing, 81, 2019.

Yue, F.-J., Li, S.-L., Liu, C.-Q., Zhao, Z.-Q., and Hu, J.: Using dual isotopes to evaluate sources and transformation of nitrogen in the Liao River, northeast China, Appl. Geochem., 36, 1-9, https://doi.org/10.1016/j.apgeochem.2013.06.009, 2013.

---

## Author Response (AR1)

**Reply to the Editor comment:**

**Editor:** Dear Authors kindly find attached the reviewer's comments. Kindly revise the manuscript and submit the annotated version along with point by point response to the comments. Kindly contact us for more information if required.

**Author:** We revised the manuscript carefully and have implemented all suggestions of the referees and the short comment. We reconsidered the title, and we changed the title into "A nitrate budget of the Bohai Sea based on an isotope mass balance model". We also made some changes on our budget and added the sensitivity analyses to section 4.3.3.

**Reply to the comments of Referee #1**

(**RC**: Referee Comment; **AR**: Author's Responds; **black** page and line numbers are related to the submitted manuscript, while **blue** page and line numbers are related to changes in the revised manuscript)

Thank you very much for thoroughly reviewing our manuscript again and for the helpful comments and suggestions that helped us to improve our manuscript. Below, you will find our responses to your comments and a description of the changes made in the revised manuscript.

**RC:** Overall the topic is of interest and fits well to the scope of the journal Biogeosciences. Moreover, a budget using the LOICZ approach was constructed using measurements but mainly data from other studies. The budget balances sources and sinks of nitrate and is supported by stable isotope data of nitrate. As a central conclusion the overwhelming role of nitrification as the major source of nitrate is presented.

**AR:** We thank the referee for his/her ideas on improving the manuscript and appreciate the careful review. In our opinion, the reviewer may not have appreciated the power of combined mass- and isotope budgets, which add a completely new dimension to standard mass-only budgets due to the process-specific isotope fractionation in the nitrogen cycle. This and an apparently poor choice of title on our side may have led to misunderstanding, which we hope to have remedied in the revision. We address this point and others raised by the referee below.

**RC:** The most significant error is the definition of equations (1) and (2). Sources and sinks are listed and supposed to be balanced. However, the source terms list nitrification and the loss terms list sedimentation. Both of these do not fulfill the criteria of a source or sink, respectively.

Nitrification is neither a source nor a sink for nitrate but simply a microbial process that converts ammonium via nitrite to nitrate. Nitrification does not generate new DIN for a system simply because the substrate of the nitrification process is ammonium and comes from internal turnover processes of organic matter. The LOICZ report (no 5 LOICZ BIOGEOCHEMICAL Modelling Guidelines, 1996) states "The important point to note with this reaction (nitrification) is that carbon and phosphorus are not directly involved in the net reaction. Again this makes the point that the relationship between NO3 and NH4 may be considered an "internal cycle" which need not be dealt with directly."

**AR:** We are sorry that our approach is apparently open to misunderstanding, and indeed changed the title of our manuscript to "A nitrate budget of the Bohai Sea based on an isotope mass balance model" in order to make our focus on reactive nitrogen clearer. Because anthropogenic impacts on biogeochemical cycles of marginal seas is always seen in amplified inputs of reactive N, we focus on this cycle. The budget in the manuscript thus is basically a budget of nitrate ion in the water mass, neither organic particles nor the sediment are included as active compartments. Instead, we expand the mass-based budget with an isotope-based budget to employ the added possibilities of dual nitrate isotopes to quantify nitrate sources and sinks. That is why nitrification and sedimentation are considered as sources and sinks, because they affect the isotope budget.

The "LOICZ approach" that the reviewer refers to links water- and salt-balance to construct carbon, nitrogen and phosphorus budget models and is an established methodology to standardize mass flux estimates of these biogeochemically important elements in coastal systems on regional to local scales (D.C. Gordon et al., 1996; Smith et al., 2005). The underlying box model approach diagnoses water, salt and CNP-fluxes, for example to decide if systems are autotrophic (production exceeds respiration) or heterotrophic (respiration exceeds production) on the basis of deviations from stoichiometric Redfield ratios. We do not aim to decide if Bohai Sea nutrient cycles indicate net autotrophy or net heterotrophy, and thus specifically do not aim to include the carbon balance associated with the LOICZ approach. We establish a more generic box model approach specifically of the reactive nitrogen pools to provide mass flux estimates of inputs and outputs to the nitrate pool. Our approach thus goes beyond a mass flux estimate (as done in previous LOICZ-type budgets for Bohai Sea, e.g., Zhang et al. (2004)) by constraining some of the branches of reactive nitrogen in this coastal sea based on the tell-tale changes in dual isotope composition of nitrate, and the dual isotopic properties of sources and sinks. This approach thus differs

fundamentally from the solely mass-balance approaches that have been done previously and that have large uncertainties specifically in the internal nitrogen turnover. In our opinion the approach taken in our study is significantly more specific and is diagnostic of several important pathways of reactive nitrogen.

Nitrification is indeed an "internal" source, as the referee says: "nitrification does not generate new DIN for a system simply because the substrate of the nitrification process is ammonium and comes from internal turnover processes of organic matter", the initial source of newly nitrified nitrate is ammonium or organic matter which releases ammonium. However, nitrification affects the stable isotopic ratios and therefore, has to be included into our budget. For our study, N in organic matter is not involved in our box model because it is simply not present as nitrate ions.

**RC:** Sedimentation is also a problematic variable for a balanced budget because it is not a removal process for nitrate. Organic material in sediments is prone to remineralization and resuspension. Only if that organic matter is permanently buried it can be considered a long-term sink. The finding that nitrification in the Bohai Bay is a very active process – generating large quantities of nitrate - is not a surprise but rather typical for coastal eutrophied systems.

**AR:** We fully agree with the referee that "only if that organic matter is permanently buried it can be considered a long-term sink". However, sedimentation in our model is defined as the nitrate removal from the water pool: because the process converts the nitrate ion to particulate nitrogen (PN), it is a sink for inorganic N and importantly, it also is associated with isotope fractionation. We only need to consider the net amount of newly produced PN, namely the amount of consumed nitrate in our model.

The significance of nitrification may be unsurprising to the reviewer, but in terms of environmental management and legislation, it makes a lot of difference whether a nitrate "target threshold concentration" can be controlled by reducing river or atmospheric inputs, or whether the nitrate is generated by sources that cannot be controlled (as is the case in Bohai Sea nitrification – the largest nitrate source).

**RC:** Another problem I have with this manuscript is that it actually consists of two independent stories; one is the field study of nitrogen compounds and stable isotopes in the Bohai Sea and the other the budget. Both are rather disconnected although the authors try to include some measurements in the budget. The field data are rather distracting from the main scope because they suggest that part of the budget is based on measurements although, most data are taken from other publications. Of course the authors write very clearly where the numbers for the budget are from and which underlying assumptions were applied to derive mean values. Nevertheless, the field data and budget remain two different stories.

**AR:** The main objective of this manuscript is the budget of mass fluxes constrained by the isotope budget, and indeed both field data and literature data are used for completing the budget. The basic data are obtained through our investigations, including the basic hydrology data (salinity and temperature), nutrients and nitrate isotopes, the properties of the particles and sediments. They are the bulk of the data used here and constitute the backbone of our study. They cannot be excluded if the whole budget needs to be constrained.

**RC:** But data from only two seasons can hardly be used for a budget averaging annual mean fluxes.

**AR:** Using data from two seasons is not ideal to extrapolate to the annual situation, but our data set brackets the intra-annual variability in the Bohai Sea. That marginal sea and the eastern Chinese seas in general are dominated by the monsoon circulation that imposes characteristic end-member states in summer and winter driven by opposite directions of monsoon. During our cruises, the early spring pattern was that of the winter season, as reflected by the vertically mixed water column, and results of the second seasonal sampling images a typical summer situation when sea water temperature was quite high and biological activity has consumed most of the nutrients. The annual situation thus is represented by the two most typical seasons. Likewise, Yellow River, the most important external riverine source of nitrate, was monitored over 5 months in order to register the dry and flood seasons.

**RC:** The results of the HAMSON model were not used at all and in the discussion the field data are only briefly mentioned.

**AR:** The HAMSOM model is described in the method part, and as stated in the text we use the modeled water transports only to calculate the "net export" of nitrate from Bohai Sea to the Yellow Sea in our mass- and isotope balance model. Because our main purpose was to estimate transports and not to delve into details of hydrodynamic circulation, we only give a brief text and do not see the point of expanding the manuscript.

**RC:** The final major concern is the lacking error estimate of the budget. All field data are subject to some degree of major or minor inaccuracy, which is not analyzed and not included in the budget calculations. Point 4.3.3 is insufficient and only addresses single sources. What's needed is an error propagation estimate.

**AR:** The errors coming from the measurement uncertainties have been done in more detail, and we report the numbers in our revision. The errors possibly incurred from fluxes differing from our preferred budget in terms of masses and isotope values are now given as a new Table 2 and a section added in the manuscript (Section 4.3.3).

**The specific reply for the small issues.**

L33/L33
    RC:    Reactive nitrogen is different to fixed nitrogen. While the first term summarises all bioavailable forms of nitrogen the latter is dedicated to diazotrophs.
    AR:    Thank you and we rephrased it to "Reactive ($N_r$) is an essential nutrient of life on earth".

L52/L52
    RC:    what is a "dramatic" increase of N/P ratios?
    AR:    The N/P ratio increased by about 30 times, we now give this number.

L61/L61
    RC:    it should be avoided to merge the process of nitrification into budget considerations.
    AR:    As explained above, we were trying to constrain a model of nitrate in the water of Bohai Sea, and even though nitrification is an internal cycling process, it leaves a significant imprint on the isotope balance.

L71/L72
    RC:    If a microbial process like nitrification is a major scope of a study it should have been measured during the field work. Including these rates could improve the study significantly.
    AR:    The nitrification rates in both water column and sediments of Chinese marginal seas are not well documented, and our manuscript remedies that lack of information. We fully agree with referee's suggestion that one way to estimate nitrification the direct measurements of nitrification by incubation experiments (Ward, 2011). A second widely used approach (Wankel et al., 2007; DiFiore et al., 2009; Sigman et al., 2009) is mass- and isotope-based modeling that we use here.

L103/L103
    RC:    are the detection limits indeed as reported? They seem high to me.
    AR:    Thank you for catching this error: the detection limit for $NO_x$ is 0.05 µmol $L^{-1}$

L133/L134
    RC:    the model has a depth resolution of 1.5m, the field data seem to have a spacing of 5-10m. This mismatch should be solved as the model validation can hardly be done with the data gathered.
    AR:    The upper 50 m of the HAMSOM model are resolved by layers of 5 m thickness. So, this coincides nicely with the resolution of our observational data. HAMSOM has been frequently applied in the Bohai Sea since last century and we consider the HAMSOM model to have been sufficiently validated in the Bohai Sea (Jia and Chen, 2021; Hainbucher et al., 2004; Huang et al., 1999).

L145/L149
    RC:    The authors may not use two seasons only to extrapolate to an entire year. Here the data of other studies could be used to generate a full annual data coverage.
    AR:    As mentioned above, the data from two seasons is not ideal to present the annual situation, but as we discuss in the manuscript, the Bohai Sea or even the eastern Chinese seas are monsoon-driven systems where most different seasons are summer and winter with opposite directions of

monsoon. The isotope data are crucial for our approach and other literature data of nitrate isotopes have to our knowledge not been published in Bohai Sea so far (see also response to RC2).

**L159, L177/L163, L181**

RC: Fig 2 and 4 have blanks. How are does ODV generate these? Are the gradients of riparian data too large?

AR: ODV users can display their data in gridded form with the calculation method called "weighted-average gridding". Users can choose the extension scale of each data point, so that the blanks are places without interpolated values. By increasing the horizontal or vertical range of each datum the blanks can be filled if necessary, but the blanks do in our opinion not obscure the patterns.

**L166/L170**

RC: Here I do not agree. The nutrient concentrations in spring are highly variable from 15-5 micromol $L^{-1}$.

AR: Here we rephrased to: "Nutrient concentrations in spring were almost vertically uniform, consistent with temperatures and salinities, and no distinct nutricline was observed".

**L168/L172**

RC: Fig 4 and 5 do not present any phosphate concentrations – the reference to the figures is not correct

AR: Nitrate or dissolved inorganic nitrogen is our key point, so that we decided against showing phosphate profiles. We deleted the reference to figures.

**L170/L174**

RC: average concentrations of all stations and depth have been calculated. Although I understand why this is done it makes of course no sense when a thermal stratification, a clear river plume and other features exist. Rephrasing and explaining this would help.

AR: We fully realize that heterogeneous distribution of any parameters introduces errors. Thus, the patterns were briefly described and displayed as graphs to inform the reader.

**L230/L237**

RC: Sv unit should use superscript

AR: We changed "1 Sv=$10^6$ m$^3$/s" to "1 Sv=$10^6$ m$^3$ s$^{-1}$".

**L244/L252**

RC: there is a typo r=. . .

AR: We changed "r=0-.78" to "r=-0.78".

**L251/L258**

RC: trace amounts are usually much lower than 0.5 micromol per liter which is the detection limit given.

AR: Thanks again for noticing this typo, it is 0.05 μmol $L^{-1}$ (L103) which is closer to the "trace amount".

**L270/L276**

RC: the assumption of similar nitrate fluxes in rivers without data based on the regional vicinity seems doubtful to me. Is the land use similar too?

AR: In the manuscript, the nitrate concentrations of Shuangtaizi River, Daling River, and Xiaoling River were set to be the same as the Daliao River. Shuantaizi River and Daliao River are quite near each other, their drainage basins are similarly populated and industrialized. Daling River and Xiaoling River are also close and both polluted by human activities (Wang et al., 2010). The nitrate concentration of these four rivers was reported to be similar (Zhang et al., 2007). Although all are quite polluted rivers, their water discharges are relatively small and only account for 4.1% of the discharge by the 8 biggest rivers into the Bohai Sea. Our estimate of nitrate concentrations for the three rivers is thus justified and erroneous estimates would only introduce small errors.

**L275/L282**

RC: Point 4.2.2 this paragraph tries to explain away all uncertainties and assumptions but the potential error is likely very high. As said above – this and the other fluxes need to be treated using error estimates.

AR: We revised the nitrate mass flux estimate of submarine fresh groundwater to a more reliable number in the revision. A new section has been added (4.3.3) that assesses the effects of uncertainties in mass flux estimates and isotope values adopted for sources and sinks.

L294/L299

RC: Point 4.2.3 was indeed the atmospheric deposition of entire China used? There should be tremendous differences across the country. May be I am misunderstanding something, but it seems that regional deposition data should be used. And again the uncertainty in the estimate needs to be included.

AR: We realize that the estimates may be problematic as well, but the results calculated by using different field data agree well and we are not aware of better data. The measurement-based estimates in the Bohai Sea are $3.4 \times 10^9$ mol year$^{-1}$ (Zhang et al., 2004) and $3.1 \times 10^9$ mol year$^{-1}$ (Liu et al., 2003), respectively. Other indirect estimates amount to $3.7 \times 10^9$ mol year$^{-1}$ (Kim et al., 2019) and $3.6 \times 10^9$ mol year$^{-1}$ (Zhao et al., 2017), respectively. We decided to use the number $3.4 \times 10^9$ mol year$^{-1}$ because it was measured directly and is in the middle range of these estimates. We discussed the impact of uncertainties of both mass flux of atmospheric deposited nitrate and dual isotopes of it in the new section 4.3.3.

L326/L329

RC: unit

AR: We adjusted the mass flux of denitrification in the revision after comparing with different newly published and global values in the literature, and discuss their effects on the budget.

L353/L357

RC: Page 18 and 19 the concerns explained above would need consideration to construct the budget differently

AR: We hope to have made clear above that we calculate the budget of "nitrate" instead of all reactive nitrogen.

L363/L367

RC: is not a hypothesis but a well known fact

AR: We changed the text "The sources of nitrate for BHS are river inputs, submarine fresh groundwater input, atmospheric deposition, and remineralization.".

L456/L486

RC: delta as Greek letter. What about sediment resuspension and transport? Wouldn't that also blur any isotope signature?

AR: The delta was corrected as Greek letter. Our data show that the $\delta^{15}N$ values of sediment increase from the Yellow River estuary to the Bohai Strait along the pathway of water and particle transport (4.55–5.58‰) and mirror a decrease in terrestrial particles with increasing distance from the estuary. Importantly, the terrestrial signal of Yellow River particles disappears in short distance from the estuary. Furthermore, the isotope data suggest that resuspension of sediment with terrestrial signature is not significant.

L463/L493

RC: Point 5 the conclusion would need a revision

AR: We hope that the explanations and the added uncertainty estimates given above are sufficient to support our conclusions, which are unchanged.

**Reply to the comments of Referee #2**

(**RC**: Referee Comment; **AR**: Author's Responds; **black** page and line numbers are related to the submitted manuscript, while blue page and line numbers are related to changes in the revised manuscript)

Thank you very much for thoroughly reviewing our manuscript again and for the helpful comments and suggestions that helped us to improve our manuscript. Below, you will find our responses to your comments and a description of the changes made in the revised manuscript.

**RC:** I think this study would need a little bit more detail discussing of the model uncertainties. There could be some uncertainties in this isotope mass balance mode due to many assumptions in this study. For example, there are many assumptions for using the end member of sedimentation (section 4.2.5). As the isotope fractionation associated with the processes of assimilation and nitrification is complicated, I think it may not be suitable to give fixed values of δ15N and δ18O to the correlated end members. I suggest to give varying values of δ15N and δ18O with reasonable range when applying to the isotope mass balance mode.

**AR:** The reviewer is of course correct in pointing out the possible errors arising from adopting fixed end member values. In the revision, we set up the uncertainties for the end members in the budget, added a table and briefly discuss the effects of adopting different endmember values in terms of masses and isotopic composition (new Section 4.3.3).

**RC:** In addition, in summer, nitrate was almost depleted in the most area of the Bohai Sea, suggesting an enhanced photosynthesis rate and assimilation rate in this season. The residual nitrate would have high δ15N and δ18O values. It may need to evaluate rationality by adopting average values of nitrate concentrations, δ15N and δ18O in the two seasons when applying to the isotope mass balance model.

**AR:** We implemented this suggestion in the revision as supplement (Supplement 4). As mentioned in the manuscript, only a subset of samples could be analyzed due to the low nitrate concentrations in summer, and most of these are from the Yellow River Diluted Water that had $[NO_3^-]$ >1.7μmol/L. The average values of $δ^{15}N$ and $δ^{18}O$ of the Bohai Sea in summer were 9.9±3.5‰ (n=23) and 8.7±3.3‰ (n=23). Although no measurements are available that could better constrain the seasonal range of nitrate isotope values, the lacking isotope data can be roughly estimated:

According to the T-S patten in summer, the Bohai Sea water can be considered as a two-end member mixture between fresh water discharged from Yellow River (YR) and sea water of central Bohai Sea, the nitrate concentration only affected by physically mixing hence can be calculated (see supplement below). The isotope effect of assimilation for nitrate in the Bohai Sea follows the "steady-state model" rather than the Rayleigh model because the Yellow River supplies nitrate continuously (Sigman and Fripiat, 2019). Thus, the estimated dual nitrate isotope values can be calculated according to equation (1) and (2):

$$δ^{15}N_{reactant} = δ^{15}N_{initial} + {}^{15}ε(1 - f) \ (1)$$

$$δ^{18}O_{reactant} = δ^{18}O_{initial} + {}^{18}ε(1 - f) \ (2)$$

In Eq.1 and Eq. 2, $f$ is equal to the observed nitrate concentration divided by of result of the two-end member model, $δ^{15}N_{initrial}$ is equal to the end member of YR, and $δ^{15}N_{reactant}$ is the estimated value of the residual nitrate, the value we need. The average of $^{15}ε$ and $^{18}ε$ adopted here are 5‰ (Granger et al., 2010; DiFiore et al., 2009; Liu et al., 2017; Wu et al., 2019; Umezawa et al., 2013; Wang et al., 2016).

The readjusted values of $δ^{15}N$ and $δ^{18}O$ for the Bohai Sea in summer is 12.8±2.7‰ (n=85) and 9.1±1.9‰ (n=85), respectively, resulting in seasonally averaged values of $δ^{15}N$ and $δ^{18}O$ of 10.3‰ and 10.6‰, respectively. These values induce about -36% to 21% deviations of the mass fluxes in our reference box model. Because this estimate is also based on the two-end member mixing model and isotopic fractionation equations, we think that this part probably is better placed in the uncertainty discussion that is included in the revision.

**Supplement: The estimate of two end member mixing of nitrate**

The YR provides warm, fresh and nitrate enriched water whereas cold, saline and nitrate depleted water was observed near the area of the outer Liaodong Bay in both spring and summer. Thus, there were two end members to be considered in a mixing model. One should be aware that a contribution of atmospheric nitrogen is included in the marine end member as well.

[Figure]

Fig. S1 Temperature vs. salinity in Bohai Sea in spring (left) and summer (right). The values adopted for the two nitrate end members were mainly based on this pattern

The values of these two end members are shown in Table S-1. The summer basic pattern of temperature and salinity was similar to that of spring. Thus, the fraction of water originating from YR and the BHS during the mixing process can be calculated follow (1) and (2):

$$S = S_r \times f_r + S_s \times f_s \ (1)$$

$$f_r + f_s = 1 \ (2)$$

where S, $S_r$ and $S_s$ refers to the observed salinity in study area, the end member value of river and sea, respectively. $f_r$ and $f_s$ refers to the fraction of river and sea water, respectively. The modeled nitrate concentration and modeled $\delta^{15}N$ and $\delta^{18}O$ values can be calculated following equations (3), (4) and (5):

$$[NO_3^-]_m = [NO_3^-]_r \times f_r + [NO_3^-]_s \times f_s \ (3)$$

$$\delta^{15}N_m[NO_3^-]_m = \delta^{15}N_r[NO_3^-]_r + \delta^{15}N_s[NO_3^-]_s \ (4)$$

$$\delta^{18}O_m[NO_3^-]_m = \delta^{18}O_r[NO_3^-]_r + \delta^{18}O_s[NO_3^-]_s \ (5)$$

where $[NO_3^-]_m$, $[NO_3^-]_r$ and $[NO_3^-]_r$ refers to the modeled nitrate concentration and the end member nitrate concentration values of river and sea, respectively. $\delta^{15}N_m/\delta^{18}O_m$, $\delta^{15}N_r/\delta^{18}O_r$ and $\delta^{15}N_s/\delta^{18}O_s$ refer to the modeled $\delta^{15}N$ and $\delta^{18}O$ values, and the end member $\delta^{15}N$ and $\delta^{18}O$ values of river and sea, respectively.

Table S1 Two end member values in Bohai Sea

| Seasons | Parameters | Riverine | Marine |
|---------|-----------|----------|--------|
| spring | Salinity | 29.9 | 33.0 |
| | Nitrate/μmol/L | 31.1 | 6.0 |
| | $\delta^{15}N‰$ | 9.5 | 6.0 |
| | $\delta^{18}O‰$ | 6.8 | 12.5 |
| summer | Salinity | 28.5 | 32.5 |
| | Nitrate/μmol/L | 13.6 | 2.0 |
| | $\delta^{15}N‰$ | 9.9 | 9.5 |
| | $\delta^{18}O‰$ | 5.3 | 8.2 |

**Reply to the comments of Short Comment #1**

(**SC**: Short Comment; **AR**: Author's Responds; **black** page and line numbers are related to the submitted manuscript, while **blue** page and line numbers are related to changes in the revised manuscript)

Thank you very much for thoroughly reviewing our manuscript and for the helpful comments and suggestions that helped us to improve our manuscript. Below, you will find our responses to your comments and a description of the changes made in the revised manuscript.

L13/L13

| | | |
|---|---|---|
| SC: | In line 13, 'The Bohai Sea' seems to be appeared first time, It is suggested to use 'The Bohai Sea (BHS)' instead here. |
| AR: | Thank you, we added (BHS) here. |

L15/L15

| | |
|---|---|
| SC: | The author is suggested to check several sentences which is difficult to read such as In line 15, 'It is therefore crucial to quantify the reactive nitrogen input to the BHS and to understand the processes and determine the quantities of nitrogen eliminated in and exported from the BHS' is suggested to revise as 'Therefore, it is crucial to quantify the reactive nitrogen input to the BHS and understand the processes and determine the quantities of nitrogen eliminated in and exported from the BHS.'. |
| AR: | Thanks, we have corrected it as suggested. |

L44-45, L642/L44-45, L679

| | |
|---|---|
| SC: | The author is suggested to check the whole manuscript about some small mistakes, such as in line 44-45 Chen, 2009 should be Chen et al., 2009, Su, 2001 could not find in the reference list. |
| AR: | We checked the "Chen, 2009" reference again and the sole author of this paper is Chen-Tung Arthur Chen (please see the link here: https://www.sciencedirect.com/science/article/abs/pii/S0924796309000748 ). The reference "Su, 2001" is called up in L642/L679 and the citation had been added. Also, the paper has been thoroughly checked for small mistakes. |

L24-25/L24-25

| | |
|---|---|
| SC: | In line 24-25, In here, Ground water should be groundwater, it is suggested that "submarine discharge of nitrate with fresh ground water' changed to 'submarine fresh groundwater discharge of nitrate'. |
| AR: | We have corrected it. |

L23-26/L24-26

| | |
|---|---|
| SC: | In line 23-26, 'The main nitrogen sources are rivers contributing 17.5%-20.6% and the combined terrestrial runoff (including submarine discharge of nitrate with fresh ground water) accounting for 22.6%-26.5% of the nitrate input to the BHS while atmospheric input contributes only 6.3%-7.4% to total nitrate.' In here, firstly you discussed about nitrogen sources, then mentioned about nitrate percentage. It seems a little confuse, please use nitrate or nitrogen (DTN? ) instead. |
| AR: | In the revision, we corrected "The main nitrogen sources" in line 23 to "The main nitrate sources" to be more explicit. |

L29-30/L29-30

| | |
|---|---|
| SC: | In line 29-30, the sentence' A further eutrophication of the BHS could, however, induce water column hypoxia and denitrification as already observed – often seasonally off river mouths - in other marginal seas.' is hard to read, please revise it as more simple way. |
| AR: | The restructured last sentence of the abstract now is: "However, a further eutrophication of the BHS could induce water column hypoxia and denitrification, as is increasingly observed in other marginal seas and seasonally off river mouths". |

L43/L42

| | |
|---|---|
| SC: | In line 43 (Smith et al., 2003; Liu et al., 2009). Please make sure if a space is needed between two citations. |
| AR: | Spaces have been inserted between multiple citations in the revised version of the manuscript. |

L70/L72
- SC: In line 70, a comma is needed after study. For this study, we analyzed. . .. . .
- AR: The comma has been added as suggested.

L73/L74
- SC: In line 73, Aim of the study - The aim of the study.
- AR: It has been corrected.

L75/L76
- SC: In line 75, . . .et al., 2011), Please remove the comma here
- AR: The comma has been removed.

L92
- SC: In line 92, The author described as 'Samples were taken monthly in May, July to November from Yellow River, and in November from Daliao River, Hai River, Luan River and Xiaoqing River (Fig. 1).', Why the water sample is only taken monthly from summer to winter in Yellow river(Also why it is not taken in June?), the other river is only a winter sample, Because it contains dry and wet season in the research region, is it enough to calibrate/validate the mass balance model using one month data?
- AR: The sampling for Yellow River (YR) was fit into the time schedule of the lead author, who participated in a ship sampling expedition to the Yellow Sea in June.

  In our view, the data still are representative for the following reasons: The flood season of the YR is July to October (MWR, 2019), so that we have 4 months representative of the flood season (July, August, September and October) and 2 months for dry season (May and November). This means that our data set covers flood and dry seasons, although admittedly not in an ideal way. YR emptied $333.8 \times 10^9$ m$^3$ water and $8.0 \times 10^9$ mol nitrate into the Bohai Sea in 2018 and accounted for 85% and 84% of water and nitrate discharge among the largest rivers discharging to the Bohai Sea, respectively. The nitrate $\delta^{15}$N and $\delta^{18}$O of YR changed little during the sampling period. Because the average values of nitrate $\delta^{15}$N and $\delta^{18}$O for rivers are mass weighted instead of arithmetic mean values, the change of nitrate $\delta^{15}$N and $\delta^{18}$O for rivers with low nitrate discharges would induce little change of the average values.

  For instance, the nitrate $\delta^{15}$N and $\delta^{18}$O of Daliao River in the flood season was reported as 20.1‰ and 9.4‰, respectively (Yue et al., 2013). Assuming that the flood season in Daliao River basin is also 4 months like in the YR basin, the flood and dry season mass weighted averaged nitrate $\delta^{15}$N and $\delta^{18}$O is 13.4‰ and 3.5‰, respectively. Combing these values with the rest of the rivers, the Daliao discharge resulted in quite small relative deviations (0.7% for $\delta^{15}$N and 0.8% for $\delta^{18}$O) to our data used in the manuscript ($\delta^{15}N_r$=10.0‰ and $\delta^{18}O_r$=1.3‰). We fully agree with the referee that a more complete monthly sampling for rivers will improve the results, but we also consider the data at hand to be quite reliable.

L135-140/L134-135, L144
- SC: In line 135-140, the author described the model by using HAMSOM, and calculate the model in year 2018. How about the warm up periods of the model? And how about the calibration/validation process, the author is suggest to describe the model more detaily.
- AR: The spin-up period of this model is 1 year. The HAMSOM model has been applied to investigate the Bohai Sea physical circulation for several decades now and has been extensively validated in the Bohai Sea (Jia and Chen, 2021; Hainbucher et al., 2004; Huang et al., 1999). This information has been added to the manuscript.

L165-176/L170-180
- SC: In line 165-176, Please uniform the nutrient name in this part, Such as there are NH4+ in the text but NH4+ -N in the figure.
- AR: $NH_4^+$ is used throughout now, other terms have been uniformed as well.

L247/L255
- SC: In line 247, 'nitrate- rich' please remove the space before rich.
- AR: The space had been removed.

**L250/L255**

SC: In line 250, '(= halo- and nutricline)' I did not understand the expression here, Could the author explain it more clearly?

AR: We changes the phrasing to "In summer, the water is stratified with the thermocline at about 8 m water depth and coinciding with halo- and nutriclines".

**L245/L250**

SC: In line 245, the author described as 'the YR is one of the major sources of these nutrients in the BHS' but not discussed the nutrient contents from other rivers, The author is suggested to described more detaily here.

AR: As we described above, Yellow River discharged $333.8 \times 10^9$ m$^3$ water and $8.0 \times 10^9$ mol nitrate to the Bohai Sea in 2018, accounting for 85% and 84% of water and nitrate discharge of all large rivers in the Bohai Sea, respectively. We added this information to the revised version.

**L249/L256**

SC: In line 249, '(see the discussion of chapter 4.2.5).' I am not sure if it is ok to refer as this way. Because it makes the reader more confused about the discussion part.

AR: The parentheses and the text included have been deleted.

**L314/L319**

SC: In line 314, 'sea water.' In manuscript, there are two descriptions as 'sea water' and 'seawater', please uniform the callings

AR: We use "seawater" in the revised version of the manuscript.

**L317/L322**

SC: In line 317, 'north China Plain' should be 'North China Plain'

AR: The "north" had been corrected to "North".

**L327**

SC: In line 327, 'The difference of the' of – between.

AR: This paragraph has been completely revised and this sentence has been removed.

**L340/L344**

SC: In line 340, 'from sediment' to 'from the sediment'.

AR: It has been corrected as suggested.

**L345/L349**

SC: In line 345, 'Sinking particles in the BHS have a $\delta^{15}$N of 5.2‰ ($\delta^{15}$N$_{sink}$),' I am confused about this part, is this data measured from this curies? This suspended particulate matter value is not shown in the manuscript (in Line 200-206, it shows the average $\delta^{15}$N of SPM in spring was 4.8±0.9‰), The author is suggested to add reference or method of this data.

AR: The $\delta^{15}$N of particles in spring and summer was 4.8±0.9‰ (L205/L213) and 5.7±0.8‰ (L207/L215), respectively. The annually averaged value $\delta^{15}$N$_{sink}$ were calculated as the mean value of spring and summer. This is described in the revised version in L349.

**L363-365/L366-368**

SC: In line 363-365, 'ground water' should be 'groundwater' 'Most important sinks' should be 'The most important sinks' 'steady state' should be 'steady-state'.

AR: It has been corrected.

**L371/L375**

SC: In line 371, 'deposited nitrate,,' please remove a comma.

AR: The extra comma has been removed.

**L440/L444**

SC: In line 440, 'in the eastern Hainan Island' remove 'the'..

AR: The extra "the" has been removed.

**References:**

D.C. Gordon, J., Boudreau, P. R., Mann, K. H., Ong, J.-E., Silvert, W. L., Smith, S. V., Wattayakorn, G., Wulff, F., and Yanagi, T.: LOICZ BIOGEOCHEMICAL MODELLING GUIDELINES, LOICZ REPORTS & STUDIES, 5, 1996.

DiFiore, P. J., Sigman, D. M., and Dunbar, R. B.: Upper ocean nitrogen fluxes in the Polar Antarctic Zone: Constraints from the nitrogen and oxygen isotopes of nitrate, Geochem. Geophys. Geosyst., 10, https://doi.org/10.1029/2009GC002468, 2009.

Granger, J., Sigman, D. M., Rohde, M., Maldonado, M., and Tortell, P.: N and O isotope effects during nitrate assimilation by unicellular prokaryotic and eukaryotic plankton cultures, Geochim. Cosmochim. Acta, 74, 1030-1040, https://doi.org/10.1016/j.gca.2009.10.044, 2010.

Hainbucher, D., Hao, W., Pohlmann, T., Sündermann, J., and Feng, S.: Variability of the Bohai Sea circulation based on model calculations, J. Mar. Syst., 44, 153-174, https://doi.org/10.1016/j.jmarsys.2003.09.008, 2004.

Huang, D., Su, J., and Backhaus, J. O.: Modelling the seasonal thermal stratification and baroclinic circulation in the Bohai Sea, Cont. Shelf Res., 19, 1485-1505, https://doi.org/10.1016/S0278-4343(99)00026-6, 1999.

Jia, B., and Chen, X. e.: Application of an ice-ocean coupled model to Bohai Sea ice simulation, Journal of Oceanology and Limnology, 39, 1-13, 10.1007/s00343-020-9168-8, 2021.

Kim, H., Park, G.-H., Lee, S.-E., Kim, Y.-i., Lee, K., Kim, Y.-H., and Kim, T.-W.: Stable isotope ratio of atmospheric and seawater nitrate in the East Sea in the northwestern Pacific ocean, Mar. Pollut. Bull., 149, 110610, https://doi.org/10.1016/j.marpolbul.2019.110610, 2019.

Liu, S. M., Zhang, J., and Jiang, W. S.: Pore water nutrient regeneration in shallow coastal Bohai Sea, China, J. Oceanogr., 59, 377-385, https://doi.org/10.1023/A:1025576212927, 2003.

Liu, S. M., Altabet, M. A., Zhao, L., Larkum, J., Song, G. D., Zhang, G. L., Jin, H., and Han, L. J.: Tracing Nitrogen Biogeochemistry During the Beginning of a Spring Phytoplankton Bloom in the Yellow Sea Using Coupled Nitrate Nitrogen and Oxygen Isotope Ratios, J. Geophys. Res.-Biogeo., 122, 2490-2508, 10.1002/2016jg003752, 2017.

MWR, C.: China River Sediment Bulletin 2018 (in Chinese), Beijing, 81, 2019.

Sigman, D. M., DiFiore, P. J., Hain, M. P., Deutsch, C., Wang, Y., Karl, D. M., Knapp, A. N., Lehmann, M. F., and Pantoja, S.: The dual isotopes of deep nitrate as a constraint on the cycle and budget of oceanic fixed nitrogen, Deep-Sea Res. Pt. I, 56, 1419-1439, https://doi.org/10.1016/j.dsr.2009.04.007, 2009.

Sigman, D. M., and Fripiat, F.: Nitrogen Isotopes in the Ocean, in: Encyclopedia of Ocean Sciences (Third Edition), edited by: Cochran, J. K., Bokuniewicz, H. J., and Yager, P. L., Academic Press, Oxford, 263-278, 2019.

Smith, S. V., Buddemeier, R. W., Wulff, F., Swaney, D. P., Camacho-Ibar, V. F., David, L. T., Dupra, V. C., Kleypas, J. A., San Diego-McGlone, M. L., McLaughlin, C., and Sandhei, P.: C, N, P Fluxes in the Coastal Zone, in: Coastal Fluxes in the Anthropocene: The Land-Ocean Interactions in the Coastal Zone Project of the International Geosphere-Biosphere Programme, edited by: Crossland, C. J., Kremer, H. H., Lindeboom, H. J., Marshall Crossland, J. I., and Le Tissier, M. D. A., Springer Berlin Heidelberg, Berlin, Heidelberg, 95-143, 2005.

Umezawa, Y., Yamaguchi, A., Ishizaka, J., Hasegawa, T., Yoshimizu, C., Tayasu, I., Yoshimura, H., Morii, Y., Aoshima, T., and Yamawaki, N.: Seasonal shifts in the contributions of the Changjiang River and the Kuroshio Current to nitrate dynamics at the continental shelf of the northern East China Sea based on a nitrate dual isotopic composition approach, Biogeosci. Disc., 10, 10143-10188, https://doi.org/10.5194/bg-11-1297-2014, 2013.

Wang, H., Li, Z., Lei, K., and Zhang, Z.: Analysis of Driving Forces and Variations of the Runoff and Suspended Sediment Discharge of the Daling and Xiaoling Rivers into the Sea over the Past Twenty Years, Research of Environmental Sciences (in Chinese), 23, 1236-1242, 2010.

Wang, W., Yu, Z., Song, X., Wu, Z., Yuan, Y., Zhou, P., and Cao, X.: The effect of Kuroshio Current on nitrate dynamics in the southern East China Sea revealed by nitrate isotopic composition, J. Geophys. Res.-Oeans, 121, 7073-7087, https://doi.org/10.1002/2016JC011882, 2016.

Wankel, S. D., Kendall, C., Pennington, J. T., Chavez, F. P., and Paytan, A.: Nitrification in the euphotic zone as evidenced by nitrate dual isotopic composition: Observations from Monterey Bay, California, Global Biogeochem. Cycles, 21, https://doi.org/10.1029/2006GB002723, 2007.

Ward, B. B.: Chapter thirteen - Measurement and Distribution of Nitrification Rates in the Oceans, in: Methods Enzymol., edited by: Klotz, M. G., Academic Press, 307-323, 2011.

Wu, Z., Yu, Z., Song, X., Wang, W., Zhou, P., Cao, X., and Yuan, Y.: Key nitrogen biogeochemical processes in the South Yellow Sea revealed by dual stable isotopes of nitrate, Estuar. Coast. Shelf Sci., 225, 106222, https://doi.org/10.1016/j.ecss.2019.05.004, 2019.

Yue, F.-J., Li, S.-L., Liu, C.-Q., Zhao, Z.-Q., and Hu, J.: Using dual isotopes to evaluate sources and transformation of nitrogen in the Liao River, northeast China, Appl. Geochem., 36, 1-9, https://doi.org/10.1016/j.apgeochem.2013.06.009, 2013.

Zhang, J., Yu, Z., Raabe, T., Liu, S., Starke, A., Zou, L., Gao, H., and Brockmann, U.: Dynamics of inorganic nutrient species in the Bohai seawaters, J. Mar. Syst., 44, 189-212, https://doi.org/10.1016/j.jmarsys.2003.09.010, 2004.

Zhang, J., XIa, B., Gui, Z., and Jiang, C.: Contaminative Conditions Evaluation of Sixteen Main Rivers Flowing into Sea Around Bohai Sea, in Summer of 2005, Environmental Science (in Chinese), 28, 2409-2415, 2007.

Zhao, Y., Zhang, L., Chen, Y., Liu, X., Xu, W., Pan, Y., and Duan, L.: Atmospheric nitrogen deposition to China: A model analysis on nitrogen budget and critical load exceedance, Atmos. Environ., 153, 32-40, https://doi.org/10.1016/j.atmosenv.2017.01.018, 2017.

---

## Referee Report (RR1)

**"A nitrate budget of the Bohai Sea based on an isotope mass balance model" by Tian, S. et al.**

The present study, by Tian, S. and colleagues, upgrades the "Reactive Nitrogen ($N_r$)" budget in the Bohai Sea (BHS) by characterizing their sources, sinks and internal nitrogen cycling. The researchers achieved their goal by employing mass-based calculations accompanied by studying dual stable-isotopes of $\delta^{15}N$ and $\delta^{18}O$ of nitrate, $\delta^{15}N$ of suspended particulate matter, sediments in depth and pre-existing information available from BHS and the surrounding riverine systems. Nitrification was found to be the key player of nitrogen cycling and massive sink by sedimentation restrict the supply of Nr to the adjacent yellow sea. The work carried out is impressive and will significantly improve the knowledge of biogeochemistry of $N_r$ in this region. The authors, addressed previous reviewer's doubts meticulously and improved in the manuscript version2. Overall, the manuscript (bg-2020-471-manuscript-version2) is clear and easy to follow. However, I suggest minor technical revision which will further improve the scientific understanding of the study performed as well the quality of the manuscript.

**Specific comments have been mentioned below:**

Line 18 (Abstract): Introduce symbol of Nitrate ($NO_3^-$).

Line 33 (Introduction): Missing "Nitrogen" for $N_r$. Rephrase the sentence without repeating dinitrogen.

Line 42: Explain DIN as dissolved inorganic nitrogen here

Line 47: Remove 'the' from fifty years

Line 48: Use only DIN here

Line 53: Remove 'the' from present day

Line 81: Replace 'by' with 'on board'

Line 92-92: Mention year.

Line 102: Add symbol of phosphate

Line 106: replace 'with' by using 'following/employing'. Please also mention the samples used for dual isotope of $NO_3^-$ analysis were filtered or not. Are those same samples used for nutrient analysis?

Line 111: Use the latest classification code "IAEA-NO-3"

Line 124: Rephrase the part "and the standard deviation is less than 0.03 ‰".

Line 161: mg $L^{-1}$ for spring values. There are various arguments on oxygen threshold for denitrification process. Please cite references for the value (0.15 mg $L^{-1}$) considered here in the manuscript.

Line 165, Figure 2 and 3: Remove '(psu)'

Line 170: Please explain exceptional prominent plume of low $NO_3^-$ and other nutrients along stations B45 and B63.

Line 176-180: Nutrient distribution in spring should be stated first and then summer as per the figures.

Figure 4 and Figure 5 both in revised manuscript represent results from Section 1. Please change Figure 5.

Line 209: replace "anti-correlated" with "inversely correlated".

Line 225: replae 'tendency' with 'trend'.

Line 249: Cite reference for spring as the season of biological productivity/Chlorophyll concentrations were not measured during present study.

Supplementary information: Line 6-8: Contradictory to the Figure S1, negative AOU values are above the thermocline, could be due to active biological production. Positive significant AOU below the thermocline represents active respiration